# Power Mean Estimation in Stochastic Monte-Carlo Tree Search

**Tuan Dam**[1]      **Odalric-Ambrym Maillard**[1]      **Emilie Kaufmann**[1]

[1]Univ. Lille, Inria, CNRS, Centrale Lille, UMR 9198-CRIStAL, F-59000 Lille, France

## Abstract

Monte-Carlo Tree Search (MCTS) is a widely-used strategy for online planning that combines Monte-Carlo sampling with forward tree search. Its success relies on the Upper Confidence bound for Trees (UCT) algorithm, an extension of the UCB method for multi-arm bandits. However, the theoretical foundation of UCT is incomplete due to an error in the logarithmic bonus term for action selection, leading to the development of Fixed-Depth-MCTS with a polynomial exploration bonus to balance exploration and exploitation [Shah et al., 2022]. Both UCT and Fixed-Depth-MCTS suffer from biased value estimation: the weighted sum underestimates the optimal value, while the maximum valuation overestimates it [Coulom, 2006]. The power mean estimator offers a balanced solution, lying between the average and maximum values. Power-UCT [Dam et al., 2019] incorporates this estimator for more accurate value estimates but its theoretical analysis remains incomplete. This paper introduces Stochastic-Power-UCT, an MCTS algorithm using the power mean estimator and tailored for stochastic MDPs. We analyze its polynomial convergence in estimating root node values and show that it shares the same convergence rate of $\mathcal{O}(n^{-1/2})$, with $n$ is the number of visited trajectories, as Fixed-Depth-MCTS, with the latter being a special case of the former. Our theoretical results are validated with empirical tests across various stochastic MDP environments.

## 1 INTRODUCTION

Monte-Carlo Tree Search (MCTS) is a family of dynamic planning algorithms that integrates asymmetric tree search and reinforcement learning (RL) to solve decision problems.

Recent advances in coupling MCTS with deep learning techniques for value estimation have facilitated the solution of complex problems with high branching factors that were considered impossible just a few years ago [Silver et al., 2016, 2017, Schrittwieser et al., 2020]. The core of the success of MCTS lies in the use of adaptive exploration of the tree using, e.g. strategies inspired by the multi-armed bandit literature. One of the most well-known algorithms is Upper Confidence bound for Trees (UCT) [Kocsis et al., 2006], which turns the UCB1 algorithm [Auer et al., 2002] into a strategy for selecting actions during tree expansion.

Kocsis et al. [2006] offers a theoretical analysis of UCT in deterministic environments establishing the convergence in selecting the optimal action for a given state for a sufficient number of simulations. However, as pointed out recently [Shah et al., 2020], there are some issues in the proof of this assertion. The problem comes from the use of a "logarithmic" bonus term within UCT, designed to balance exploration and exploitation during tree-based search. This approach is built upon the assumption that the concentration of regret for the underlying recursively dependent nonstationary MABs will exponentially converge to its expected value as the number of steps advances. However, as demonstrated by Audibert et al. [2009], the validity of this assumption is doubtful, given that the underlying regret converges polynomially rather than exponentially.

Building upon these insights, Shah et al. [2022] propose an adapted version of UCT incorporating a polynomial bonus term instead of the logarithmic bonus term in UCT. They offer a comprehensive theoretical analysis to show that the resulting algorithm, called Fixed-depth-MCTS, ensures polynomial convergence in value function estimation at the root node. However, their work is mostly focused on deterministic environments.[1] Moreover, the Power-UCT algorithm, introduced by Dam et al. [2019] as a generalization of UCT

---

[1]The first version of their work [Shah et al., 2020] mentioned the stochastic case as an open question, while it is treated in Appendix A of the journal version [Shah et al., 2022]. However, they use some high-level reduction argument, and the stochastic version

similarly focuses on deterministic environment and its current analysis suffers from the same shortcoming as that of UCT [Kocsis et al., 2006].

This paper introduces a novel MCTS algorithm for Stochastic MDPs using power mean for the value estimator, called Stochastic-Power-UCT. We propose the same form of a polynomial bonus term as introduced in the work of Shah et al. [2020]. We show that Stochastic-Power-UCT also ensures the polynomial concentration of value estimation at the root node. We complement our method by empirically performing a variety of experiments in stochastic tasks confirming our theory. Thus, our *contribution* is threefold:

(i) We propose Stochastic-Power-UCT with a complete theoretical convergence guarantee using the power mean backup operator in stochastic MCTS.

(ii) We demonstrate that the estimated value function at the root node of our tree converges polynomially to the optimal value, exhibiting the same convergence rate as Fixed-depth-MCTS [Shah et al., 2020], which is $\mathcal{O}(n^{-1/2})$, with $n$ is the number of visited trajectories. Our method employs power mean as value estimators, with the average mean utilized in Fixed-depth-MCTS being a specific case.

(iii) We conduct various experiments in SyntheticTree toy task and in various stochastic MDPs, which support our theoretical analysis.

## 2   SETTING

In this section, we first provide some background knowledge about Markov Decision Processes, and then we give an overview of Monte-Carlo Tree Search.

**Markov Decision Process**   In Reinforcement Learning (RL), the agent aims to make optimal decisions in an environment modeled as a Markov Decision Process (MDP), a standard framework for sequential decision-making. We focus on a discrete-time discounted MDP, defined as $\mathcal{M} = \langle \mathcal{S}, \mathcal{A}, \mathcal{R}, \mathcal{P}, \gamma \rangle$, where $\mathcal{S}$ is the state space, $\mathcal{A}$ is the finite action space, $\mathcal{R} : \mathcal{S} \times \mathcal{A} \times \mathcal{S} \to \mathbb{R}$ is the reward function, $\mathcal{P} : \mathcal{S} \times \mathcal{A} \to \mathcal{S}$ is the transition dynamics, $\gamma \in [0, 1)$ is the discount factor. A policy $\pi \in \Pi : \mathcal{S} \to \mathcal{A}$ represents a probability distribution over feasible actions given the current state.

We denote $Q^\pi(s, a)$ as a $Q$ value function under the policy $\pi$ defined based on Bellman equation as

$$Q^\pi(s, a) \triangleq \sum_{s'} \mathcal{P}(s'|s, a) \left[ \mathcal{R}(s, a, s') + \gamma \sum_{a'} \pi(a'|s') Q(s', a') \right],$$

and the value function under the policy $\pi$ is denoted as $V^\pi(s) = \max_{a \in \mathcal{A}} Q^\pi(s, a)$. Our goal is to find the optimal policy that maximizes the value function at each state,

of their algorithm is not explicitly presented.

where the optimal value function at state $s$ is defined as $V^\star(s) = \max_{a \in \mathcal{A}} Q^\star(s, a)$, with $Q^\star(s, a)$ the optimal $Q$ value function at state $s$ action $a$ satisfies the optimal Bellman equation [Bellman, 1954]

$$Q^*(s, a) \triangleq \sum_{s'} \mathcal{P}(s'|s, a) \left[ \mathcal{R}(s, a, s') + \gamma \max_{a'} Q^*(s', a') \right].$$

**Monte-Carlo Tree Search**   Monte-Carlo Tree Search (MCTS) (see Browne et al. [2012] for a survey) is a family of online planning strategies that combines Monte-Carlo sampling with forward tree search to find on-the-fly optimal decisions. MCTS algorithms use a black-box model of the environment in simulation to build a planning tree. An MCTS algorithm consists of four components: Selecting nodes to traverse in the tree based on the current statistical information, expanding the tree, evaluating the leaf that has been reached (using possibly a roll-out in the environments), and using the collected rewards from the environment along the chosen path to update the algorithm. The key elements influencing the quality of a particular algorithms are an effective value update operator and an efficient node selection strategy in the tree.

**Formalization**   An MCTS algorithm adaptively collects trajectories in an MDP, starting from an initial state $s_0$, to build a planning tree. Each trajectory continues until it either reaches a leaf node or a node at some maximum depth $H$. At the end of each trajectory, a playout policy (which can be either deterministic or stochastic) is applied from the last node reached, to provide an evaluation of the corresponding state. After $t$ trajectories, it may output two things:

- $\widehat{a}_t$, a guess for the best action to take in state $s_0$
- $\widehat{V}_t(s_0)$ an estimator of the optimal value in $s_0$

Its quality performance can be evaluated by its convergence rate $r(t)$, of the form

$$\mathbb{E}\left[V^\star(s_0) - Q^\star(s_0, \widehat{a}_t)\right] \leq r(t) \tag{1}$$

$$\text{or } \left| \mathbb{E}\left[V^\star(s_0) - \widehat{V}_t(s_0)\right] \right| \leq r(t). \tag{2}$$

We shall analyze an MCTS algorithm using some maximal planning horizon $H$ and a playout policy $\pi_0$ with value $V_0$. Denoting by $s_h$ a node at depth $h$ in the tree (that is identified to some state that may be reached in $h$ steps from the root note), we can define inductively $\widetilde{V}(s_H) = V_0(s_H)$ and, for all $h \leq H - 1$,

$$\widetilde{Q}(s_h, a) = r(s_h, a) + \gamma \sum_{s_{h+1} \in \mathcal{A}_s} \mathcal{P}(s_{h+1}|s_h, a)\widetilde{V}(s_{h+1}),$$

$$\widetilde{V}(s_h) = \max_a \widetilde{Q}(s_h, a),$$

where $r(s_h, a)$ is the mean of intermediate reward at state $s_h$ after taking action $a$. Then we have $|Q^\star(s_0, a) - \widetilde{Q}(s_0, a)| \leq \gamma^H \|V^\star - V_0\|_\infty$ (actually the supremum could be restricted

to all states reachable in $H$ steps from $s_0$). The purpose of an MCTS algorithm is to minimize the convergence rate $r(t)$ by building an estimate of $\widetilde{Q}(s_0, a)$ and $\widetilde{V}(s_0)$ in order to be finally able to estimate $Q^\star(s_0, a)$ and the best action in the root note $a_\star = \arg\max_a Q^\star(s_0, a)$.

We are now ready to present our particular MCTS instance, Stochastic-Power-UCT algorithm.

## 3 STOCHASTIC POWER-UCT

In this section, we first present a generic UCT like algorithm and then we present our Stochastic-Power-UCT algorithm.

### 3.1 GENERIC UCT-LIKE ALGORITHM

For each node $s_h$ in depth $h$ of the search tree, and for each available action $a \in \mathcal{A}_{s_h}$, we denote by

- $\widehat{V}_t(s_h)$ the value estimate built after $s_h$ has been visited $t$ times at depth $h$
- $\widehat{Q}_t(s_h, a)$ the Q-value estimate built after $(s_h, a)$ has been visited $t$ times at depth $h$

We denote by $T_{s_h}(t)$, $T_{s_h,a}(t)$ and $T_{s_h,a}^{s_{h+1}}(t)$ the number of visits of $s_h$ (respectively $(s_h, a)$, $(s_h, a, s_{h+1})$) in depth $h$ after $t - 1$ MCTS trajectories.

A generic UCT-like algorithm depends on a sequence of bonus functions $B(t, s_h, a)$ for each depth $h$. It sequentially plays trajectories from a starting state $s_0$ until some leaf of the search tree or some maximal depth $H$ is reached (nodes at this depth are also called leaves). At the leaves, the playout policy $\pi_0$ is used to provide a (possibly random) evaluation of the value $V_0$. We assume that repeated calls to the playout policy in a state $s$ provides i.i.d. samples from a distribution with mean $V_0(s)$. If the playout is not stochastic but provided by a neural network, then the distribution is a Dirac delta centered at $V_0(s)$. The playout could also be fully stochastic (i.e., outputting a sum of discounted rewards under $\pi_0$ starting from $s$) or a mix of both.

The $t$-th trajectory collected is

$$\{s_0^t = s_0, a_0, r_0, s_1, a_1, r_1, s_2, a_2, r_2, \ldots, s_{\ell_t}, \widetilde{V}(s_{\ell_t})\},$$

where $\ell_t \leq H$ is the length of the trajectory and $\widetilde{V}(s_{\ell_t})$ is the value of the playout performed in $s_{\ell_t}$. For each $h \leq \ell_t$:

$$a_h = \underset{a \in \mathcal{A}_{s_h}}{\arg\max} \left\{ \widehat{Q}_{T_{s_h,a}(t)}(s_h, a) + B\left(t, s_h, a\right) \right\},$$

and $s_{h+1} \sim \mathcal{P}(\cdot | s_h, a_h)$. If $s_{\ell_t}$ is a leaf of the search tree with $\ell_t < H$, we add to the search tree all the $Q$-nodes $(s_{\ell_t}, a)$ for all available actions in $s_{\ell_t}$. After a trajectory is collected, the number of visits of the state (resp. state-action

pairs) in the trajectory are updated, and the corresponding values (resp. Q-values) estimates are computed.

After $t$ trajectories, the guess $\widehat{a}_t$ will be

$$\widehat{a}_t = \underset{a \in \mathcal{A}_{s_0}}{\arg\max} \widehat{Q}_{T_{s_0,a}(t)}(s_0, a),$$

and the estimate of the value of the root will be $\widehat{V}_t(s_0)$, where $\widehat{V}_t$ is a value operator to be specified.

### 3.2 STOCHASTIC POWER-UCT

A UCT-like algorithm is fully characterized by:

- the definition of value and Q-value estimates
- the choice of the bonus function $B(t, s_h, a)$
- the maximal depth $H$ and playout policy

In the vanilla UCT algorithm [Kocsis and Szepesvári, 2006], $B(t, s_h, a) = C\sqrt{\frac{\log(T_{s_h}(t))}{T_{s_h,a}(t)}}$ with $C$ is an exploration constant, which is the bonus used by the UCB algorithm, used to select action in a stochastic bandit algorithm. Other bonusses have been used in practice too [Browne et al., 2012] and we know since the work of Shah et al. [2022] that these logarithmic bonus are not sufficient to prove convergence. In our Stochastic-Power-UCT algorithm, we define the sequence of bonus function as

$$B(t, s_h, a) = C \frac{T_{s_h}(t)^{\frac{b_{h+1}}{\beta_{h+1}}}}{T_{s_h,a}(t)^{\frac{\alpha_{h+1}}{\beta_{h+1}}}}, h = 0, 1, \ldots, H - 1,$$

where along the tree from depth $0$ to depth $H$ we maintain $\{b_i\}_{i=0}^H$, $\{\alpha_i\}_{i=0}^H$, $\{\beta_i\}_{i=0}^H$ as algorithmic constants satisfy conditions as in Table 1, and dividing by zero assume to be $+\infty$.

Particular choices for the sequences of parameter constants presented above have been proposed based on the theoretical study, which will be described in the next section. We highlight that these choices are the same as the Fixed-Depth-MCTS algorithm from Shah et al. [2020] for $1 \leq p \leq 2$ and when $p > 2$, an extra condition $0 < \alpha_i - \frac{\beta_i}{p} < 1$ is needed. Furthermore, the Fixed-Depth-MCTS algorithm Shah et al. [2020] has been studied for deterministic settings, while our method is proposed for stochastic environments with general power mean value estimators.

As the the Values and Q-values estimates, they are the average of the sum of discounted rewards starting from this state (resp. state-action) obtained in all past trajectories going through this state (resp. state-action). They can be also be computed inductively as follows. If $s$ is a leaf of the search tree at depth $h$, $\widehat{V}_t(s)$ is the average of $t$ playout obtained

**Input:** root node state $s_0$
**Output:** optimal action at the root node

```
R = Rollout (s, depth)
    Ṽ(s) = average of the call to π₀(s)
    return Ṽ(s)

a = SelectAction (sₕ, depth = h, greedy=false, t)
    if greedy == false then
        a = arg max { Q̂_{T_{sₕ,a}(t)}(sₕ,a) + C (T_{sₕ}(t)^{b_{h+1}/β_{h+1}}) / (T_{sₕ,a}(t)^{α_{h+1}/β_{h+1}}) }
         a
    else
        a = arg max { Q̂_{T_{sₕ,a}(t)}(sₕ,a) }
         a
    return a

SimulateV (sₕ, depth, t)
    a ← SelectAction(sₕ, depth =h, greedy = false, t)
    SimulateQ (sₕ, a, depth =h, t)
    T_{sₕ}(t) ← T_{sₕ}(t) + 1
    V̂_{T_{sₕ}(t)}(sₕ) ← ( Σₐ (T_{sₕ,a}(t)/T_{sₕ}(t)) (Q̂_{T_{sₕ,a}(t)})^p (sₕ,a) )^{1/p}
```

```
SimulateQ (sₕ, a, depth = h, t)
    s_{h+1} ~ P(·|sₕ,a)
    r(sₕ,a) ~ R(sₕ, a, s_{h+1})
    if s_{h+1} ∉ Terminal then
        if Node s_{h+1} not expanded then
            V̂_{T_{s_{h+1}}(t)}(s_{h+1}) = Rollout(s_{h+1}, depth)
        else
            SimulateV (s_{h+1}, depth = h + 1, t)
    Q̂_{T_{sₕ,a}(t)}(sₕ,a)
    ← (Q̂_{T_{sₕ,a}(t)}(sₕ,a)T_{sₕ,a}(t)+r(sₕ,a)+γV̂_{T_{s_{h+1}}(t)}(s_{h+1})) / (T_{sₕ,a}(t)+1)
    T_{sₕ,a}(t) ← T_{sₕ,a}(t) + 1

MainLoop
    t = 0
    while t ≤ n do
        SimulateV (s₀, depth = 0, t)
        t ← t + 1
    return SelectAction (s₀, greedy = true, n)
```

**Algorithm 1:** Stochastic-Power-UCT with $\gamma$ is a discount factor. $n$ : is the number of rollouts. $\{b_i\}_{i=0}^H$, $\{\alpha_i\}_{i=0}^H$, $\{\beta_i\}_{i=0}^H$ are positive algorithmic constants that satisfy conditions as in Table 1. $\pi_0$ is a rollout policy. $C$ is an exploration constant.

Table 1: Conditions for algorithmic constants. $i \in [0, H]$.

| Algorithmic constants, each row as AND conditions |
| --- |
| $b_i < \alpha_i; b_i > 2.$ |
| $1 \le p \le 2; \alpha_i \le \frac{\beta_i}{2}$ OR $p > 2; \alpha_i \le \frac{\beta_i}{2}; 0 < \alpha_i - \frac{\beta_i}{p} < 1.$ |
| $\alpha_i \left(1 - \frac{b_i}{\alpha_i}\right) \le b_i < \alpha_i.$ |
| $\alpha_i = (b_{i+1} - 1)\left(1 - \frac{b_{i+1}}{\alpha_{i+1}}\right).$ |
| $\beta_i = (b_{i+1} - 1).$ |

by using the playout policy[2]. For internal nodes, we define inductively, for all $t$,

$$\widehat{V}_t(s_h) = \left( \sum_{a \in \mathcal{A}_{s_h}} \frac{T_{s_h,a}(t)}{t} \left( \widehat{Q}_{T_{s_h,a}(t)}(s_h,a) \right)^p \right)^{\frac{1}{p}}, \quad (3)$$

$$\widehat{Q}_t(s_h, a) = \frac{1}{t} \sum_{i=1}^{t} \left[ r^i(s_h, a) + \gamma \widehat{V}_{T^{s_{h+1}}_{s_h,a}(i)}(s_{h+1}) \right], \quad (4)$$

where $p \in [1, +\infty)$. We denote $r^i(s_h, a)$ is the $i$-th instantaneous reward collected after visiting $(s_h, a)$ in depth $h$. $T^{s_{h+1}}_{s_h,a}(i)$ is the number of visits of $(s_h, a)$ to $s_{h+1}$ after timestep $i$. Detailed can be found at Algorithm 1.

**Remark 1.** *We described above the practical implementation of MCTS algorithm, for which sometimes the maximal depth $H$ is sometimes even set to $+\infty$. For the theoretical analysis however, the maximal depth $H$ will be crucial and*

[2]note that the playout policy will be called several times only in leaves that are at depth $H$

*we will actually analyze a variant of this algorithm that always collects trajectories of length $H$.*

# 4 THEORETICAL ANALYSIS

Planning in MCTS requires a sequence of decisions along the tree, with each internal node acting as a non-stationary bandit. The empirical mean at these nodes shifts due to the action selection strategy. To address this problem, we first analyze non-stationary multi-armed bandit settings, focusing on the concentration properties of the power-mean backup for each arm compared to the optimal value. We then apply these findings to MCTS.

## 4.1 NON-STATIONARY POWER MEAN MULTI-ARMED BANDIT

We consider a class of non-stationary multi-armed bandit (MAB) problems. Let us consider $K \ge 1$ arms or actions of interest. Let $X_{a,t}$ denote the random reward obtained by playing arm $a \in [K]$ at the time step $t$, the reward is bounded in $[0, R]$. $\widehat{\mu}_{a,n} = \frac{1}{n}\sum_{t=1}^{n} X_{a,t}$ is the average reward collected at arm $a$ after n times. Let $\mu_{a,n} = \mathbb{E}[\widehat{\mu}_{a,n}]$. We define

**Definition 1.** *A sequence of estimators $(\widehat{V}_n)_{n \ge 1}$ concentrates at rate $\alpha, \beta$ towards some limit $V$ under certain conditions on $\alpha, \beta$ if there exists a constant $c > 0$ such that the following property holds:*

$$\forall n \ge 1, \forall \varepsilon > n^{-\frac{\alpha}{\beta}}, \mathbb{P}\left( |\widehat{V}_n - V| > \varepsilon \right) \le cn^{-\alpha}\varepsilon^{-\beta}.$$

*We write $\widehat{V}_n \xrightarrow[n\to\infty]{\alpha,\beta} V$.*

We assume that the reward sequence $\{X_{a,t}\}$ is a non-stationary process satisfying the following assumption:

**Assumption 1.** *Consider $K$ arms that for $a \in [K]$, let $(\widehat{\mu}_{a,n})_{n\geq 1}$ be a sequence of estimator satisfying*

$$\widehat{\mu}_{a,n} \xrightarrow[n\to\infty]{\alpha,\beta} \mu_a.$$

Let us define $\mu_\star = \max_{a\in[K]}\{\mu_a\}$. In our study, we assume that $\mu_\star$ is unique, and there is a strict gap between the best optimal value and the second best value. Under Assumption 1, we consider the following optimistic action selection strategy, based on the estimator $\widehat{\mu}_{a,n}$ and using a similar bonus as the one in Stochastic-Power-UCT. More precisely, the algorithm starts by selecting each arm once. Then, given $b < \alpha, b > 2, \beta > 0$, at each time step $n > K$, the selected action is

$$a_n = \underset{a\in\{1...K\}}{\arg\max}\left\{\widehat{\mu}_{a,T_a(n)} + C\frac{n^{\frac{b}{\beta}}}{T_k(n)^{\frac{\alpha}{\beta}}}\right\}, \quad (5)$$

where $T_a(n) = T_a(n) = \sum_{t=1}^{n-1}\mathbb{1}(a_t = a)$ denotes the number of selections of arm $a$ prior to round $n$. Given a constant $1 \leq p < \infty$, we define

$$\widehat{\mu}_n(p) = \left(\sum_{a=1}^{K}\frac{T_a(n)}{n}\widehat{\mu}_{a,T_a(n)}^p\right)^{\frac{1}{p}}$$

as the power mean value backup operator.

We establish the concentration properties of the average mean backup operator $\widehat{\mu}_n(p)$ towards the mean value of the optimal arm $\mu_*$, as shown in Theorem 1.

**Theorem 1.** *For $a \in [K]$, let $(\widehat{\mu}_{a,n})_{n\geq 1}$ be a sequence of estimators satisfying $\widehat{\mu}_{a,n} \xrightarrow[n\to\infty]{\alpha,\beta} \mu_a$ and let $\mu_\star = \max_a\{\mu_a\}$. Assume that the arms are sampled according to the strategy equation 5 with parameters $\alpha, \beta, b$ and $C$. Assume that $p, \alpha, \beta$ and $b$ satisfy one of these two conditions:*

*(i) $1 \leq p \leq 2$ and $\alpha \leq \frac{\beta}{2}$*

*(ii) $p > 2$ and $0 < \alpha - \frac{\beta}{p} < 1$*

*If $\alpha\left(1 - \frac{b}{\alpha}\right) \leq b < \alpha$ then the sequence of estimators $\widehat{\mu}_n(p)$ satisfies*

$$\widehat{\mu}_n(p) \xrightarrow[n\to\infty]{\alpha',\beta'} \mu_\star$$

*for $\alpha' = (b-1)\left(1 - \frac{b}{\alpha}\right)$ and $\beta' = (b-1)$ for some value of the constant $C$ in equation 5 that depends on $K, b, \alpha, p, \Delta_{\min}$ with $\Delta_{\min} = \min_{a:\mu_a < \mu_\star}(\mu_\star - \mu_a)$.*

Based upon the results of Stochastic-Power-UCT using power mean as the value backup operator on the described non-stationary multi-armed bandit problem, we derive theoretical results for Stochastic-Power-UCT in an MCTS tree.

## 4.2 MONTE-CARLO TREE SEARCH

Based on the results from the non-stationary multi-armed bandit from the last section, we can derive theoretical analysis for the Stochastic-Power-UCT in an MCTS tree where we consider each node in the tree as a Non-stationary multi-armed bandit problem.

We start with a result of the following lemma which plays an important role in the analysis of our MCTS algorithm.

**Lemma 1.** *For $m \in [M]$, let $(\widehat{V}_{m,n})_{n\geq 1}$ be a sequence of estimator satisfying $\widehat{V}_{m,n} \xrightarrow[n\to\infty]{\alpha,\beta} V_m$, and there exists a constant $L$ such that $\widehat{V}_{m,n} \leq L, \forall n \geq 1$. Let $X_i$ be an iid sequence with mean $\mu$ and $S_i$ be an iid sequence from a distribution $p = (p_1, \ldots, p_M)$ supported on $\{1, \ldots, M\}$. Introducing the random variables $N_m^n = \#|\{i \leq n : S_i = s_m\}|$, we define the sequence of estimator*

$$\widehat{Q}_n = \frac{1}{n}\sum_{i=1}^{n}X_i + \gamma\sum_{m=1}^{M}\frac{N_m^n}{n}\widehat{V}_{m,N_m^n}.$$

*Then with $2\alpha \leq \beta, \beta > 1$,*

$$\widehat{Q}_n \xrightarrow[n\to\infty]{\alpha,\beta} \mu + \sum_{m=1}^{M}p_m V_m.$$

The proof of Lemma 1 can be found in the Appendix. This result is important as it can be used to show that the Q-value estimates at a certain depth $h$ concentrate at the same rate $(\alpha, \beta)$ as the value estimates of the children nodes. Then, thanks to Theorem 1, the value estimate at depth $h$, which is computed using a power mean, will concentrate at a different rate $(\alpha', \beta')$. Proceeding by induction from depth $H$ to depth 0 allows us to derive Theorem 2, which shows the polynomial concentration of the values and Q-values at the root note. We note that this part of analysis is fairly similar to the analysis of Shah et al. [2022]. However, its two main ingredients required some innovation. Indeed, Theorem 1 is specific to our power-mean value back-up operator, while Lemma 1 is specific to the concentration of Q values in stochastic MDPs.

**Theorem 2.** *When we apply the Stochastic-Power-UCT algorithm, with $\{b_i\}_{i=0}^{H}$, $\{\alpha_i\}_{i=0}^{H}$, $\{\beta_i\}_{i=0}^{H}$ as algorithmic constants satisfying the conditions in Table 1, we have*

*(i) For any node $s_h$ at the depth $h^{th}$ in the tree ($h = [0, 1\ldots, H]$),*

$$\widehat{V}_n(s_h) \xrightarrow[n\to\infty]{\alpha_h,\beta_h} \widetilde{V}(s_h).$$

*(ii) For any node $s_h$ at the depth $h^{th}$ in the tree ($h = [0\ldots, H-1]$),*

$$\widehat{Q}_n(s_h, a) \xrightarrow[n\to\infty]{\alpha_{h+1},\beta_{h+1}} \widetilde{Q}(s_h, a), \text{ for all } a \in \mathcal{A}_{s_h}.$$

*Proof.* We will prove the Theorem by induction on the depth $H$ of the tree.

Initial step $H = 1$.

The state at the root node is $s_0$. Let us assume that $r^t(s_0, a_k)$ is the intermediate reward at time step $t$, after visiting $(s_0, a_k)$, and go to state $s_1 \sim \mathcal{P}(\cdot|s_0, a_k)$. Let us assume that $r(s_0, a_k)$ is the mean of $(s_0, a_k)$. We recall the definition of $\widetilde{Q}(s_0, a_k)$,

$$\widetilde{Q}(s_0, a_k) = r(s_0, a_k) + \gamma \sum_{s_1 \in \mathcal{A}_{s_0}} \mathcal{P}(s_1|s_0, a_k)\widetilde{V}(s_1)$$

where $\widetilde{V}(s_1)$ is the average value of the rollout policy $\pi_0$ at state $s_1$, $\mathcal{A}_{s_0}$ is the set of feasible actions at state $s_0$, $|\mathcal{A}_{s_0}| = M$, $\mathcal{P}(s_1|s_0, a_k)$ is the probability transition of taking action $a_k$ at state $s_0$ to state $s_1$.

$(i)$ satisfies for any state $s_1 \sim \mathcal{P}(\cdot|s_0, a_k)$ as

$$\widehat{V}_n(s_1) \xrightarrow[n \to \infty]{\alpha_1, \beta_1} \widetilde{V}(s_1), \tag{6}$$

because each value at the leaf node $\widehat{V}_n(s_1)$ is the average of i.i.d call to the playout policy $\pi_0(s)$.

From equation 4, we have

$$\widehat{Q}_n(s_0, a_k) = \frac{1}{n}\sum_{t=1}^{n}\left[r^t(s_0, a_k) + \gamma\widehat{V}_{T^{s_1}_{s_0, a_k}(t)}(s_1)\right]. \tag{7}$$

By applying Lemma 1 with $X_t$ is the intermediate reward $r^t(s_0, a_k)$ at time $t$, $p = (p_1, p_2, ...p_M)$ is the probability transition dynamic of taking action $a_k$ at state $s_0$. For $m \in [M]$, each $(\widehat{V}_{m,n})_{n \geq 1}$ at time step $n$ satisfies

$$\widehat{V}_{m,n}(s_1) \xrightarrow[n \to \infty]{\alpha_1, \beta_1} \widetilde{V}(s_1), \text{ with } s_1 \in \{s_m\}, m = 1, 2, 3...M,$$

where $s_m \sim \mathcal{P}(\cdot|s_0, a_k)$, we have

$$\widehat{Q}_n(s_0, a) \xrightarrow[n \to \infty]{\alpha_1, \beta_1} \widetilde{Q}(s_0, a), \text{ for all } a \in \mathcal{A}_{s_0}. \tag{8}$$

Therefore at the root node $s_0$, applying Theorem 1, with the results of (8) and because

$$\widehat{V}_n(s_0) = \left(\sum_{a \in \mathcal{A}_s} \frac{T_{s_0, a}(n)}{n}\left(\widehat{Q}_{T_{s_0, a}(n)}(s_0, a)\right)^p\right)^{\frac{1}{p}}, \tag{9}$$

$p \in [1, +\infty)$, we have

$$\widehat{V}_n(s_0) \xrightarrow[n \to \infty]{\alpha_0, \beta_0} \widetilde{V}(s_0), \tag{10}$$

with $\alpha_0, \beta_0$ satisfies conditions in Table 1. *From (6), (10), we conclude that (i) is correct when the depth of the tree is 1.*

$(ii)$ is correct according to (8).

Let us assume that the theorem holds with the tree of depth $H - 1$.

Now let us consider the tree with depth $H$.

When we take an action $a_k$ at the root node state $s_0$ and get state $s_1 \sim \mathcal{P}(\cdot|s_0, a_k)$, we go to a subtree with depth $H - 1$. According to the induction hypothesis, in the subtree with the root node $s_1$, we have with $h = [1 \ldots, H]$

$$\widehat{V}_n(s_h) \xrightarrow[n \to \infty]{\alpha_h, \beta_h} \widetilde{V}(s_h), \tag{11}$$

and with $h = [1 \ldots, H - 1]$

$$\widehat{Q}_n(s_h, a) \xrightarrow[n \to \infty]{\alpha_{h+1}, \beta_{h+1}} \widetilde{Q}(s_h, a), \text{ for all } a \in \mathcal{A}_{s_h}. \tag{12}$$

We now consider the root node at state $s_0$.

We apply again Lemma 1 with $X_t$ is the intermediate reward $r^t(s_0, a_k)$ at time $t$ and each $(\widehat{V}_{m,n})_{n \geq 1}$ at time step $n$ satisfies (because of (11))

$$\widehat{V}_{m,n}(s_1) \xrightarrow[n \to \infty]{\alpha_1, \beta_1} \widetilde{V}(s_1), \text{ with } s_1 \in \{s_m\}, m = 1, 2, 3...M,$$

where $s_m \sim \mathcal{P}(\cdot|s_0, a_k)$, we have

$$\widehat{Q}_n(s_0, a) \xrightarrow[n \to \infty]{\alpha_1, \beta_1} \widetilde{Q}(s_0, a), \text{ for all } a \in \mathcal{A}_{s_0}. \tag{13}$$

At the root node $s_0$, We apply again Theorem 1, with the concentration results of Q value at (13) and the value backup operator at root state $s_0$ (9), we have

$$\widehat{V}_n(s_0) \xrightarrow[n \to \infty]{\alpha_0, \beta_0} \widetilde{V}(s_0), \tag{14}$$

with $\alpha_0, \beta_0$ satisfies conditions in Table 1.

Combining (11) and (14) concludes for $(i)$.

Combining (12) and (13) concludes for $(ii)$.

The results of Theorem 2 hold for any node in the tree with the tree of depth $(H)$. By induction, we can conclude the proof. $\square$

Finally, we state the expected payoff of Value estimation at the root node polynomial decays, as shown below.

**Theorem 3 (Convergence of Expected Payoff).** *We have at the root node $s_0$, with the best possible parameter tuning that*

$$\left|\mathbb{E}[\widehat{V}_n(s_0)] - \widetilde{V}(s_0)\right| \leq \mathcal{O}(n^{-1/2}).$$

*Proof.* Using the convexity of $f(x) = |x|$ and applying

Jensen's inequality we have

$$\left|\mathbb{E}[\widehat{V}_n(s_0)] - \widetilde{V}(s_0)\right| \leq \mathbb{E}[\left|\widehat{V}_n(s_0)] - \widetilde{V}(s_0)\right|]$$

$$= \int_0^{+\infty} \mathbb{P}\left(\left|\widehat{V}_n(s_0) - \widetilde{V}(s_0)\right| \geq s\right) ds$$

$$\leq \int_0^{n^{-\frac{\alpha_0}{\beta_0}}} 1 ds + \int_{n^{-\frac{\alpha_0}{\beta_0}}}^{+\infty} c_0 n^{-\alpha_0} s^{-\beta_0} ds$$

$$\leq n^{-\frac{\alpha_0}{\beta_0}} + c_0 n^{-\alpha_0} \left(\frac{s^{-\beta_0+1}}{-\beta_0+1}\right)\Big|_{n^{-\frac{\alpha_0}{\beta_0}}}^{+\infty}$$

$$= (\frac{c_0}{\beta_0 - 1} + 1) n^{-\frac{\alpha_0}{\beta_0}}.$$

Because $\frac{\alpha_0}{\beta_0} \leq \frac{1}{2}$ (Theorem 1), then the best possible rate we can estimate is

$$\left|\mathbb{E}[\widehat{V}_n(s_0)] - \widetilde{V}(s_0)\right| \leq \mathcal{O}(n^{-1/2}).$$

That concludes the proof.

**Remark 2.** *These results demonstrate that both Stochastic-Power-UCT and Fixed-Depth-MCTS share the same convergence rate for value estimation at the root node, which is $\mathcal{O}(n^{-1/2})$. By selecting algorithmic constants from Table 1 such that $\frac{\alpha_i}{\beta_i} = 1/2$ and $\frac{b_i}{\beta_i} = 1/4$ for $i \in [0, H]$, we achieve the optimal rate. This choice leads us to adopt the exploration bonus:*

$$B_h(n, s, a) = C \frac{n^{1/4}}{T_{s,a}(n)^{1/2}},$$

*where $C$ represents an exploration constant. Our findings align with those of Shah et al. [2022], but our finding more broadly applies to the power mean estimator, and the average mean is a special case.*

$\square$

## 5 EXPERIMENTS

In this section, we present experimental results demonstrating the numerical advantages of Stochastic-Power-UCT compared to UCT [Kocsis et al., 2006], Power-UCT [Dam et al., 2019] and Fixed-Depth-MCTS [Shah et al., 2022] in SyntheticTree, FrozenLake ($4 \times 4$), FrozenLake ($8 \times 8$) and Taxi environments. The discount factor is set $\gamma = 1$ in SyntheticTree and $\gamma = 0.99$ in FrozenLake and Taxi environments. Hyperparameter can be found in the Appendix.

As the results from Remark 2, the exploration bonus is chosen as $C \frac{n^{1/4}}{T_{s,a}(n)^{1/2}}$ with C is an exploration constant in all environments. In SyntheticTree, we run further experiments with adaptive choice of parameters $\alpha_i, \beta_i, b_i$ for $i \in [0, H]$ satisfied Table 1 to confirm the theoretical study.

**Synthetic Tree**

We evaluate Power-UCT using the synthetic tree toy problem Dam et al. [2021]. The problem involves a tree with depth $d$ and branching factor $k$. Each edge of the tree has a random value between 0 and 1, and at each leaf, a Gaussian distribution is used as an evaluation function resembling the return of random rollouts. The mean of the Gaussian distribution is the sum of the values assigned to the edges connecting the root node to the leaf, while the standard deviation is set to a constant $\sigma$ (we set $\sigma = 0.5$ in our experiments). After trying different values, To ensure stability, the means are normalized between 0 and 1. We introduce stochasticity into the environment by altering the transition probabilities: there is a 80% chance of moving to the intended node and a 20% chance of moving to a different node with equal probability. We conduct 25 experiments on five trees with five runs each, covering all combinations of branching factors $k = \{2, 4, 6, 8, 10, 16\}$ and depths $d = \{1, 2, 3, 4\}$. We compute the value estimation error at the root node.

Fig. 1 shows the convergence of the value estimations at the root node in the Synthetic Tree environment with different settings. In detail, Fig. 1a shows the performance of Stochastic-Power-UCT with different values of $p$, where we find that $p = 2$ outperforms all other $p$ values. In Fig 1b, We compare Stochastic-Power-UCT with UCT, Power-UCT, and Fixed-Depth-MCTS. We also find that $p = 2$ works the best. In Fig 1a and Fig 1b, we choose the exploration bonus $C \frac{n^{1/4}}{T_{s,a}(n)^{1/2}}$. In Fig 1c, we use the exploration bonus $C \frac{n^{b_i/\beta_i}}{T_{s,a}(n)^{\alpha_i/\beta_i}}, i \in [0, H]$ with $\alpha_i, \beta_i, b_i$ satisfied Table 1. The convergence results of Stochastic-Power-UCT shown in Fig 1 confirm the theoretical study.

**Frozen Lake**

In the OpenAI Gym [Brockman et al., 2016], the *FrozenLake* problem presents a classic empirical MDP environment. The goal is to guide an agent through an ice-grid world, avoiding unstable spots that lead to water. The environment's stochastic nature adds challenge, as the agent moves in the intended direction only one-third of the time, and otherwise in one of two tangential directions. Reaching the target earns a reward of $+1$, while other outcomes yield zero reward. In Table. 2a and Table. 2b, Stochastic-Power-UCT ($p = 2$) outperforms UCT and Fixed-Depth-MCTS ($p = 1$) with $2^{14}, 2^{15}$ rollouts in FrozenLake $4 \times 4$, and $2^{13}, 2^{14}$ rollouts in FrozenLake $8 \times 8$. In most cases, Stochastic-Power-UCT ($p = 2$) has the average mean higher than others.

**Taxi** In the *Taxi* environment Dearden et al. [1998], agents navigate a 7x6 grid from the top left to the top right, encountering walls that block movement. Simply reaching the end yields no reward; the agent must collect three passengers scattered across the grid before reaching the target position. Rewards vary based on the number of passengers collected and delivered successfully. We

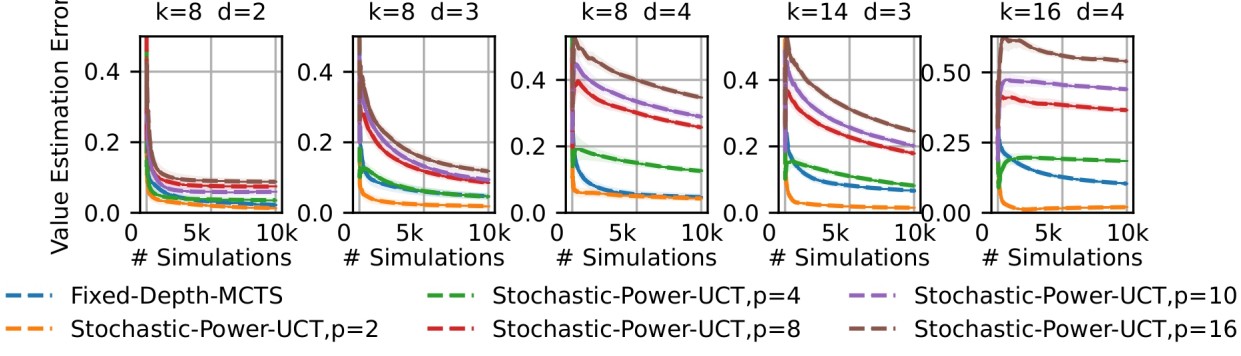

(a) Compare Stochastic-Power-UCT with difference $p$ value: $p = 1.0$(Fixed-Depth-MCTS), 2.0, 4.0, 6.0, 8.0, 10.0, 16.0. The exploration bonus is chosen as $C\frac{n^{1/4}}{T_{s,a}(n)^{1/2}}$ with C as an exploration constant.

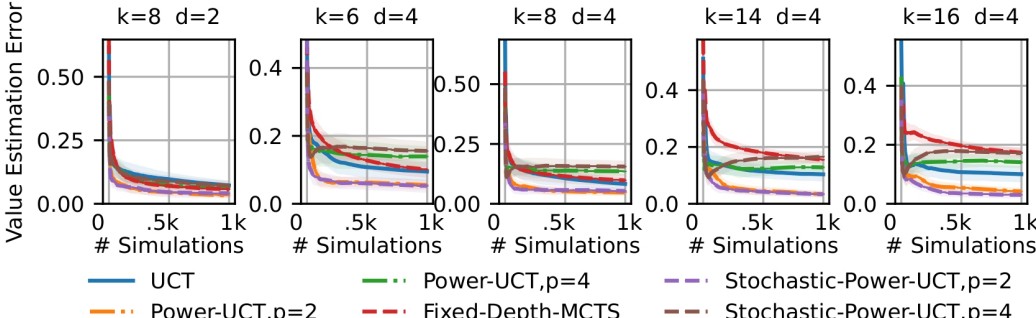

(b) Compare UCT, Power-UCT, Fixed-Depth-MCTS and Stochastic-Power-UCT. The exploration bonus is chosen as $C\frac{n^{1/4}}{T_{s,a}(n)^{1/2}}$ with C as an exploration constant.

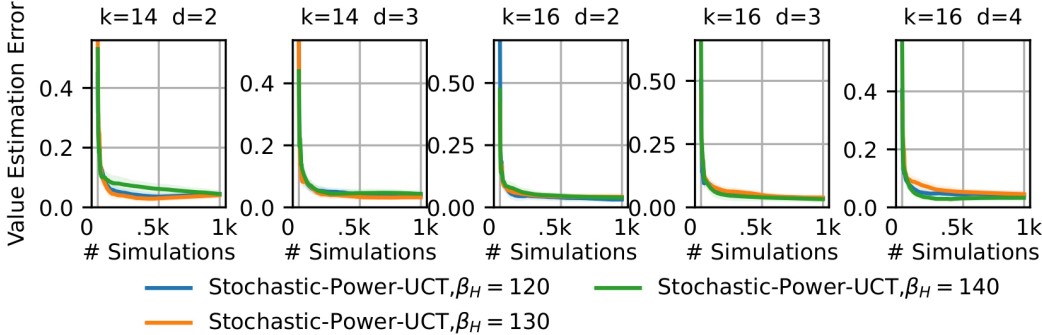

(c) Compare Stochastic-Power-UCT with the exploration bonus $C\frac{n^{\frac{b_i}{\beta_i}}}{T_k(n)^{\frac{\alpha_i}{\beta_i}}}$ where the adaptive parameters of $\{\alpha_i\}_0^H, \{\beta_i\}_0^H, \{b_i\}_0^H$ and $p$ with difference initial $\beta_H = 120, 130, 140$ value chosen accordingly satisfied Table 1.

Figure 1: We show the convergence of the value estimate at the root node to the respective optimal in Synthetic tree environment.

Table 2: Mean and two times standard deviation of discounted total reward, over 1000 evaluation runs, of UCT, Fixed-Depth-MCTS($p = 1$) and Stochastic-Power-UCT($p = 2$, and $p = 2.2$) in FrozenLake ($4 \times 4$), FrozenLake ($8 \times 8$), and Taxi environments (in Taxi, we perform 20 evaluation runs). Top row: number of simulations at each time step. Bold denotes no statistically significant difference to the highest mean (t-test, $p < 0.05$).

(a) FrozenLake 4x4

| Algorithm | 2048 | 4096 | 8192 | 16384 | 32768 | 65536 | 131072 | 262144 |
|---|---|---|---|---|---|---|---|---|
| UCT | $\mathbf{0.10 \pm 0.01}$ | $\mathbf{0.13 \pm 0.01}$ | $\mathbf{0.20 \pm 0.02}$ | $0.27 \pm 0.02$ | $0.37 \pm 0.02$ | $\mathbf{0.43 \pm 0.02}$ | $\mathbf{0.44 \pm 0.02}$ | $0.44 \pm 0.02$ |
| $p = 1$ | $\mathbf{0.11 \pm 0.01}$ | $0.15 \pm 0.02$ | $\mathbf{0.20 \pm 0.02}$ | $0.29 \pm 0.02$ | $0.35 \pm 0.02$ | $\mathbf{0.41 \pm 0.02}$ | $\mathbf{0.45 \pm 0.02}$ | $0.48 \pm 0.02$ |
| $p = 2$ | $\mathbf{0.15 \pm 0.02}$ | $\mathbf{0.21 \pm 0.02}$ | $\mathbf{0.31 \pm 0.02}$ | $\mathbf{0.37 \pm 0.02}$ | $0.39 \pm 0.02$ | $\mathbf{0.44 \pm 0.02}$ | $\mathbf{0.45 \pm 0.02}$ | $0.47 \pm 0.02$ |
| $p = 2.2$ | $\mathbf{0.16 \pm 0.02}$ | $\mathbf{0.23 \pm 0.02}$ | $\mathbf{0.30 \pm 0.02}$ | $\mathbf{0.37 \pm 0.02}$ | $0.40 \pm 0.02$ | $\mathbf{0.42 \pm 0.02}$ | $\mathbf{0.45 \pm 0.02}$ | $0.50 \pm 0.02$ |

(b) FrozenLake 8x8

| Algorithm | 1024 | 2048 | 4096 | 8192 | 16384 | 32768 | 65536 | 131072 |
|---|---|---|---|---|---|---|---|---|
| UCT | $\mathbf{0.01 \pm 0.006}$ | $\mathbf{0.02 \pm 0.007}$ | $\mathbf{0.05 \pm 0.01}$ | $0.07 \pm 0.01$ | $0.12 \pm 0.01$ | $\mathbf{0.18 \pm 0.01}$ | $\mathbf{0.22 \pm 0.01}$ | $\mathbf{0.29 \pm 0.01}$ |
| $p = 1$ | $\mathbf{0.02 \pm 0.006}$ | $0.02 \pm 0.008$ | $\mathbf{0.06 \pm 0.001}$ | $0.07 \pm 0.01$ | $0.10 \pm 0.01$ | $\mathbf{0.17 \pm 0.01}$ | $\mathbf{0.23 \pm 0.01}$ | $\mathbf{0.29 \pm 0.01}$ |
| $p = 2$ | $\mathbf{0.02 \pm 0.006}$ | $\mathbf{0.04 \pm 0.09}$ | $\mathbf{0.06 \pm 0.01}$ | $\mathbf{0.09 \pm 0.01}$ | $\mathbf{0.14 \pm 0.01}$ | $\mathbf{0.21 \pm 0.01}$ | $\mathbf{0.25 \pm 0.01}$ | $\mathbf{0.33 \pm 0.01}$ |
| $p = 2.2$ | $\mathbf{0.01 \pm 0.006}$ | $\mathbf{0.04 \pm 0.009}$ | $\mathbf{0.06 \pm 0.01}$ | $\mathbf{0.10 \pm 0.01}$ | $\mathbf{0.12 \pm 0.01}$ | $\mathbf{0.19 \pm 0.01}$ | $\mathbf{0.26 \pm 0.01}$ | $\mathbf{0.31 \pm 0.01}$ |

(c) Taxi

| Algorithm | 512 | 1024 | 2048 | 4096 | 8192 | 16384 |
|---|---|---|---|---|---|---|
| UCT | $1.03 \pm 0.68$ | $\mathbf{1.20 \pm 0.56}$ | $1.28 \pm 0.54$ | $\mathbf{1.25 \pm 0.69}$ | $\mathbf{1.32 \pm 0.54}$ | $\mathbf{1.66 \pm 0.83}$ |
| $p = 1$ | $\mathbf{0.69 \pm 0.24}$ | $1.11 \pm 0.76$ | $2.22 \pm 1.01$ | $1.63 \pm 0.82$ | $1.52 \pm 0.53$ | $1.96 \pm 1.04$ |
| $p = 2$ | $\mathbf{0.63 \pm 0.36}$ | $0.92 \pm 0.54$ | $\mathbf{1.72 \pm 0.81}$ | $\mathbf{1.49 \pm 0.64}$ | $2.24 \pm 0.83$ | $\mathbf{2.94 \pm 0.95}$ |
| $p = 2.2$ | $\mathbf{0.85 \pm 0.45}$ | $\mathbf{0.76 \pm 0.47}$ | $\mathbf{1.22 \pm 0.68}$ | $1.15 \pm 0.49$ | $\mathbf{2.45 \pm 0.90}$ | $3.07 \pm 0.98$ |

introduce stochasticity by setting 50% chance of moving to the intended direction and 50% chance of moving to other directions with equal probability. This stochastic environment necessitates thorough exploration.

As shown in Table. 2c, when we increase the number of rollouts, Stochastic-Power-UCT ($p = 2$) and Stochastic-Power-UCT ($p = 2.2$) outperforms UCT and Fixed-Depth-MCTS ($p = 1$) with $2^{13}, 2^{14}$ rollouts.

**Remark 3.** *In our experiment, we find that when $p = 2$, Stochastic-Power-UCT consistently outperforms UCT, Fixed-Depth-MCTS ($p = 1$) and outperform Stochastic-Power-UCT with other $p$ value.*

## 6 CONCLUSION

Monte Carlo tree search (MCTS) is emerging as an effective approach with many applications in games and Autonomous car driving, Robot path planning, and robot assembly tasks. However, understanding of the theoretical foundations of MCTS remains limited. In this work, we introduce Stochastic-Power-UCT, using power mean as the value estimation and a polynomial exploration bonus term, which is specifically designed for stochastic MDP scenarios. Our contribution extends to a thorough theoretical study of the convergence rate of $\mathcal{O}(n^{-1/2})$ for value estimation at the root node of Stochastic-Power-UCT. Moreover, empirical validation of our theoretical findings is performed in SyntheticTree and various stochastic MDP environments, confirming the theoretical claims of our approach. Our work

put one more step for future research efforts aimed at improving the theoretical understanding and practical applicability of MCTS in stochastic environments. One can think of extending our work by studying Power-UCT in adversarial settings. Furthermore, hybrid combination of learning in reinforcement learning and planning in MCTS could be promising with applications in robotics.

**Acknowledgments**

This work has been supported by the French Ministry of Higher Education and Research, the Hauts-de-France region, Inria, the MEL, the I-Site ULNE regarding project RPILOTE-19-004-APPRENF, the Inria A.Ex. SR4SG project, and the Inria-Kyoto University Associate Team "RELIANT".

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

# Power Mean Estimation in Monte-Carlo Tree Search (Supplementary Material)

**Tuan Dam**[1]        **Odalric-Ambrym Maillard**[1]        **Emilie Kaufmann**[1]

[1]Univ. Lille, Inria, CNRS, Centrale Lille, UMR 9198-CRIStAL, F-59000 Lille, France

## A    OUTLINE

- Notations will be described in Section B.
- Supporting Lemmas are presented in Section C.
- The Convergence of Stochastic-Power-UCT in Non-stationary multi-armed bandits is shown in Section D.
- Experimental setup and Hyperparameter selection are provided in Section E.

## B    NOTATIONS

Table 3: List of all notations for Non-stationary Multi-arms bandit.

| Notation | Type | Description |
|:---:|:---:|:---:|
| $K$ | $\mathbb{N}$ | Number of arms |
| $T_a(t)$ | $\mathbb{N}$ | Number of visitations at arm $a$ after $t$ timesteps |
| $\mu_a$ | $\mathbb{R}$ | mean value of arm $a$ |
| $a_\star$ | $\mathcal{A}$ | optimal action |
| $\mu_\star$ | $\mathbb{R}$ | mean value of an optimal arm. We assume it is unique. |
| $\widehat{\mu}_n(p)$ | $\mathbb{R}$ | power mean estimator, with a constant $p \in [1, +\infty)$ |
| $\widehat{\mu}_{a,n}$ | $\mathbb{R}$ | mean estimator of arm $a$ after $n$ visitations |

## C    SUPPORTING LEMMAS

In this section, we will present all necessary supporting Lemmas for the main theoretical analysis.

We start with a result of the following lemma which plays an important role in the analysis of our MCTS algorithm.

**Lemma 1.** *For $m \in [M]$, let $(\widehat{V}_{m,n})_{n \geq 1}$ be a sequence of estimator satisfying $\widehat{V}_{m,n} \xrightarrow[n \to \infty]{\alpha,\beta} V_m$, and there exists a constant $L$ such that $\widehat{V}_{m,n} \leq L, \forall n \geq 1$. Let $X_i$ be an iid sequence with mean $\mu$ and $S_i$ be an iid sequence from a distribution $p = (p_1, \ldots, p_M)$ supported on $\{1, \ldots, M\}$. Introducing the random variables $N_m^n = \#|\{i \leq n : S_i = s_m\}|$, we define the sequence of estimator*

$$\widehat{Q}_n = \frac{1}{n} \sum_{i=1}^{n} X_i + \gamma \sum_{m=1}^{M} \frac{N_m^n}{n} \widehat{V}_{m,N_m^n}.$$

*Then with $2\alpha \leq \beta, \beta > 1$,*

$$\widehat{Q}_n \xrightarrow[n \to \infty]{\alpha,\beta} \mu + \sum_{m=1}^{M} p_m V_m.$$

*Proof.* Let $p = (p_1, p_2, \ldots p_M), p \in \triangle^M$ where $\triangle^M = \{x \in \mathbb{R}^M : \sum_{i=1}^{M} x_i = 1, x_i \geq 0\}$ is the $(M-1)$-dimensional simplex. Without loss of generality, we assume that $p_m > 0$ for all $m$. Let us study a random vector $\widehat{p}_n = (\frac{N_1^n}{n}, \frac{N_2^n}{n}, \ldots, \frac{N_M^n}{n})$. Let us define $V = (V_1, V_2, \ldots V_M)$. Let $\widehat{X}_n = \frac{1}{n} \sum_{i=1}^{n} X_i, \widehat{V}_n = (\widehat{V}_{1,N_1^n}, \widehat{V}_{2,N_2^n}, \ldots, \widehat{V}_{M,N_M^n}), \sum_{i=1}^{M} N_i^n = n, N_i^n$ is the number of times that population $i$ was observed. We have $\widehat{Q}_n = \widehat{X}_n + \gamma \left\langle \widehat{p}_n, \widehat{V}_n \right\rangle$. Therefore,

$$\mathbb{P}\left( \widehat{Q}_n - (\mu + \gamma \langle p, V \rangle) \geq \epsilon \right) \leq \mathbb{P}\left( \widehat{X}_n - \mu \geq \frac{1}{2}\epsilon \right) + \mathbb{P}\left( \gamma \left\langle \widehat{p}_n, \widehat{V}_n \right\rangle - \gamma \langle p, Y \rangle \geq \frac{1}{2}\epsilon \right)$$

$$\leq \exp\{-2n\frac{\epsilon^2}{4}\} + \underbrace{\mathbb{P}\left( \left\langle \widehat{p}_n, \widehat{V}_n \right\rangle - \langle p, Y \rangle \geq \frac{1}{2\gamma}\epsilon \right)}_{A}.$$

To upper bound A, let us consider $\left\langle \widehat{p}_n, \widehat{V} \right\rangle - \langle p, V \rangle = \left\langle (\widehat{p}_n - p), \widehat{V}_n \right\rangle + \left\langle p, (\widehat{V} - V) \right\rangle$. Then,

$$A \leq \underbrace{\mathbb{P}\left( \left\langle (\widehat{p}_n - p), \widehat{V}_n \right\rangle \geq \frac{1}{4\gamma}\epsilon \right)}_{A_1} + \underbrace{\mathbb{P}\left( \left\langle p, (\widehat{V}_n - V) \right\rangle \geq \frac{1}{4\gamma}\epsilon \right)}_{A_2}.$$

By applying a Hölder inequality to $\widehat{p}_n - p$ and $\widehat{V}$, we obtain

$$\left\langle (\widehat{p}_n - p), \widehat{V}_n \right\rangle \leq \| \widehat{p}_n - p \|_1 \| \widehat{V}_n \|_\infty = \| \widehat{p}_n - p \|_1 L,$$

with $L \geq \| \widehat{V} \|_\infty$, $L$ is a constant. Then we can derive

$$A_1 = \mathbb{P}\left( \left\langle (\widehat{p}_n - p), \widehat{V}_n \right\rangle \geq \frac{1}{4\gamma}\epsilon \right)$$

$$\leq \mathbb{P}\left( \| \widehat{p}_n - p \|_1 L \geq \frac{1}{4\gamma}\epsilon \right)$$

$$= \mathbb{P}\left( \| \widehat{p}_n - p \|_1 \geq \frac{1}{4\gamma L}\epsilon \right).$$

According to Weissman et al. [2003], we have for any $M \geq 2$ and $\delta \in [0, 1]$

$$\mathbb{P}\left( \| \widehat{p}_n - p \|_1 \geq \sqrt{\frac{2M \ln(2/\delta)}{n}} \right) \leq \delta.$$

Define $\epsilon = \sqrt{\frac{2M \ln(2/\delta)}{n}}$, therefore $\delta = 2 \exp\{\frac{-n\epsilon^2}{2M}\}$, we have

$$\mathbb{P}\left( \| \widehat{p}_n - p \|_1 \geq \epsilon \right) \leq 2 \exp\{\frac{-n\epsilon^2}{2M}\}.$$

Therefore,

$$A_1 \le \mathbb{P}\bigg( \parallel \widehat{p}_n - p \parallel_1 \ge \epsilon \bigg) \le 2\exp\{\frac{-n\epsilon^2}{32M\gamma^2 L^2}\}.$$

We also have

$$A_2 = \mathbb{P}\bigg( \sum_{m=1}^{M} p_m (\widehat{V}_{m,N_m^n} - V_m) \ge \frac{1}{4\gamma}\epsilon \bigg)$$

$$\le \sum_{m=1}^{M} \mathbb{E}\bigg[ \mathbb{P}\bigg( \frac{1}{N_m^n} \sum_{t=1}^{N_m^n} V_{m,t} - V_m \ge \frac{1}{4\gamma p_m}\epsilon \big| N_m^n \bigg) \bigg]$$

$$\le \sum_{m=1}^{M} \mathbb{E}\bigg[ c(N_m^n)^{-\alpha}(\frac{\epsilon}{4\gamma p_m})^{-\beta} \bigg].$$

Let us define an event $\mathcal{E} = \left\{ N_m^n > \frac{np_m}{2} \right\}$. Therefore,

$$A_2 \le \sum_{m=1}^{M} \mathbb{E}\bigg[ c(\frac{np_m}{2})^{-\alpha}(\frac{\epsilon}{4\gamma p_m})^{-\beta} \bigg] + \sum_{m=1}^{M} \mathbb{E}\bigg[ \mathbb{P}(N_m^n \le \frac{np_m}{2}) \bigg]$$

$$= \sum_{m=1}^{M} (c2^{\alpha+2\beta}\gamma^\beta p_m^{-\alpha+\beta})n^{-\alpha}\epsilon^{-\beta} + \sum_{m=1}^{M} \mathbb{E}\bigg[ \mathbb{P}(N_m^n - p_m n \le -\frac{p_m n}{2}) \bigg]$$

$$\le \sum_{m=1}^{M} (c2^{\alpha+2\beta}\gamma^\beta p_m^{-\alpha+\beta})n^{-\alpha}\epsilon^{-\beta} + \sum_{m=1}^{M} \exp\bigg\{ -2n(\frac{p_m n}{2})^2 \bigg\} \tag{15}$$

Therefore,

$$A \le A_1 + A_2 \le 2\exp\{\frac{-n\epsilon^2}{32M\gamma^2 L^2}\} + \sum_{m=1}^{M} (c2^{\alpha+2\beta}\gamma^\beta p_m^{-\alpha+\beta})n^{-\alpha}\epsilon^{-\beta} + \sum_{m=1}^{M} \exp\bigg\{ -2n(\frac{p_m n}{2})^2 \bigg\}.$$

That leads to

$$\mathbb{P}\bigg( \widehat{Q}_n - (\mu + \gamma\langle p, V\rangle) \ge \epsilon \bigg) \le \exp\{-2n\frac{\epsilon^2}{4}\} + 2\exp\{\frac{-n\epsilon^2}{32M\gamma^2 L^2}\} + \sum_{m=1}^{M} (c2^{\alpha+2\beta}\gamma^\beta p_m^{-\alpha+\beta})n^{-\alpha}\epsilon^{-\beta}$$

$$+ \sum_{m=1}^{M} \exp\bigg\{ -2n(\frac{p_m n}{2})^2 \bigg\} \le c' n^{-\alpha}\epsilon^{-\beta},$$

with $c' > 0$ depends on $c, M, \alpha, \beta, p_i$. Here we need

$$2\alpha \le \beta, \tag{16}$$

to argue that $\exp(-cn\varepsilon^2) = \mathcal{O}(n^{-\alpha}\varepsilon^{-\beta})$. By following the same steps, we can derive

$$\mathbb{P}\bigg( \widehat{Q}_n - (\mu + \gamma\langle p, V\rangle) \le -\epsilon \bigg) \le c' n^{-\alpha}\epsilon^{-\beta}.$$

Therefore, with $n \ge 1, \epsilon > 0$,

$$\mathbb{P}\bigg( \Big| \widehat{Q}_n - (\mu + \gamma\langle p, V\rangle) \Big| \ge \epsilon \bigg) \le c' n^{-\alpha}\epsilon^{-\beta}. \tag{17}$$

This means

$$\widehat{Q}_n \xrightarrow[n\to\infty]{\alpha,\beta} \mu + \gamma\sum_{m=1}^{M} p_m V_m,$$

which concludes the proof. □

**Lemma 2.** *Let consider non-negative variables $x, y \in \mathbb{R}^+$, and a constant $m$ that $0 \le m \le 1$. Then*

$$(x + y)^m \le x^m + y^m. \tag{18}$$

*Proof.* With $y = 0$, or $x = 0$, the inequality (18) becomes correct. Let consider the case where $x > 0, y > 0$, the inequality (18) can be written as

$$\left(\frac{x}{y} + 1\right)^m \le \left(\frac{x}{y}\right)^m + 1.$$

Let us define a function

$$f(t) = (t + 1)^m - t^m - 1, (t > 0).$$

We can see that

$$f'(t) = m(t + 1)^{m-1} - mt^{m-1} = m\left((t + 1)^{m-1} - t^{m-1}\right) \le 0 \text{ with } m \in [0, 1], t > 0,$$

because $g(x) = x^{m-1}$ is a decreasing function with $m \in [0, 1], x > 0$. Therefore,

$$f(t) \le f(0) = 0 \text{ with } t > 0.$$

So that,

$$(t + 1)^m - t^m - 1 \le 0, (t > 0).$$

with $t = \frac{x}{y} \ge 0$, we can derive the inequality (18). $\square$

We use Minkowski's inequality as shown below

**Lemma 3(Minkowski's inequality).** *Given $p \ge 1, \{x_i, y_i\} \in \mathbb{R}, i = 1, 2, ..., n$, then we have the following inequality*

$$\left(\sum_i (|x_i + y_i|)^p\right)^{\frac{1}{p}} \le \left(\sum_i (|x_i|)^p\right)^{\frac{1}{p}} + \left(\sum_i (|y_i|)^p\right)^{\frac{1}{p}} \tag{19}$$

*Proof.* This is a basic result. $\square$

# D  CONVERGENCE OF STOCHASTIC-POWER-UCT IN NON-STATIONARY MULTI-ARMED BANDITS

In an MCTS tree, each node functions as a non-stationary multi-armed bandit, with the average mean drifting due to the action selection strategy. To address this, we first study the convergence of Stochastic-Power-UCT in non-stationary multi-armed bandits, where action selection is based on Thompson sampling, and the power mean backup operator is used at the root node. Detailed descriptions of Stochastic-Power-UCT in non-stationary bandit settings can be found in the Theoretical Analysis section of the main article.

We establish the convergence and concentration properties of the power mean backup operator in non-stationary bandits, as detailed in Theorem 1 for Stochastic-Power-UCT which mainly based on the results of Lemma 10. To derive the results for Lemma 10, we need results of Lemma 4, Lemma 6, Lemma 7, Lemma 8, and Lemma 9. These lemmas collectively support the theoretical understanding of Stochastic-Power-UCT in non-stationary multi-armed bandit settings.

Lemma 4 shows the upper bound for probability of the difference between the mean value estimation at the optimal arm (with $T_{a_*}(n)$ number of visitations) and the optimal value $\mu_\star$.

Lemma 6 show the crucial on the high-probability bound on the number of selection of each sub-optimal arm, which based on the results of Lemma 5. Lemma 7 show the upperbound for absolute value of the difference of the power mean estimator and the optimal value. Lemma 8, and Lemma 9 show intermediate results that helps to derive results of Lemma 10.

**Lemma 4.** *Consider a bandit problem defined as in Section 4.1. Let us define $A(n) = \left(\frac{2Cn^{\frac{b}{\beta}}}{\Delta}\right)^{\frac{\beta}{\alpha}}$, where $\Delta = \min_{a \in [K]} \{\mu_* - \mu_a\}, a \neq a_*$, with $R \ge \epsilon \ge n^{-\frac{\alpha}{\beta}}$ then we have*

$$\mathbb{P}\left(\left|\widehat{\mu}_{a_\star, T_{a_*}(n)} - \mu_\star\right| > \epsilon\right) \le \sum_{a \neq a_*}^K \mathbb{P}\left(T_a(n) > (A(n) + 1)\right) + \frac{c}{\alpha - 1}\epsilon^{-\beta}(n - (K - 1)A(n) + 1)^{-\alpha+1}. \tag{20}$$

*Proof.* Consider an event $\mathcal{E} \stackrel{\text{def}}{=} \left\{ \sum_{a \neq a_*}^{K} T_a(n) > (K-1)(A(n)+1) \right\}$. Then,

$$\mathbb{P}\left(\left|\widehat{\mu}_{a_\star, T_{a_*}(n)} - \mu_\star\right| > \epsilon\right) \leq \mathbb{P}\left(\sum_{a \neq a_*}^{K} T_a(n) > (K-1)(A(n)+1)\right)$$

$$+ \underbrace{\mathbb{P}\left(\sum_{a \neq a_*}^{K} T_a(n) \leq (K-1)(A(n)+1); \left|\widehat{\mu}_{a_\star, T_{a_*}(n)} - \mu_\star\right| \geq \epsilon\right)}_{D_1}. \quad (21)$$

When $\sum_{a \neq a_*}^{K} T_a(n) \leq (K-1)(A(n)+1) \Rightarrow T_{a_*}(n) = n - \sum_{a \neq a_*}^{K} T_a(n) \geq n - (K-1)(A(n)+1)$, so that with $\alpha > 0$

$$D_1 \leq \mathbb{P}\left(T_{a_*}(n) \geq n - (K-1)(A(n)+1); \left|\widehat{\mu}_{a_\star, T_{a_*}(n)} - \mu_\star\right| \geq \epsilon\right) \leq \sum_{t=n-(K-1)(A(n)+1)}^{n} \mathbb{P}\left(\left|\widehat{\mu}_{a_\star, t} - \mu_\star\right| \geq \epsilon\right)$$

$$\leq \sum_{t=n-(K-1)(A(n)+1)}^{n} ct^{-\alpha}\epsilon^{-\beta}$$

$$\leq c\epsilon^{-\beta}\left(\int_{n-(K-1)(A(n)+1)-1}^{\infty} t^{-\alpha}dt\right) = \frac{c}{\alpha-1}\epsilon^{-\beta}(n-(K-1)(A(n)+1)-1)^{-\alpha+1}(\text{ because } \alpha > 2). \quad (22)$$

Combining Equation (21) and Equation (22), we can conclude the proof. $\qquad\square$

We introduce the notation $U_{a,t,s} = \widehat{\mu}_{a,s} + C\dfrac{t^{\frac{b}{\beta}}}{s^{\frac{\alpha}{\beta}}}$ and we first borrow two lemmas of Shah et al. [2022].

Introducing for all $a$ the quantity

$$A_a(t) := \inf\left\{s \leq t : C\frac{t^{\frac{b}{\beta}}}{s^{\frac{\alpha}{\beta}}} \leq \frac{\Delta_a}{2}\right\} = \left(\frac{2C}{\Delta_a}\right)^{\frac{\beta}{\alpha}} t^{\frac{b}{\alpha}},$$

where $\Delta_a = \mu_* - \mu_a$, the concentration properties permits to prove the following Lemma.

**Lemma 5.** *Let $n \geq 1$.*

*(i) For all $s \in \{1, \ldots, n\}$, $\mathbb{P}(U_{a,n,s} < \mu_a) \leq cC^{-\beta}n^{-b}$*

*(ii) For all $s \in \{A_a(n), \ldots, n\}$, $\mathbb{P}(U_{a,n,s} > \mu_\star) \leq cC^{-\beta}n^{-b}$*

*Proof.* 1.

$$\mathbb{P}(U_{a,n,s} < \mu_a) = \mathbb{P}\left(\widehat{\mu}_{a,s} - \mu_a < -C\frac{n^{\frac{b}{\beta}}}{s^{\frac{\alpha}{\beta}}}\right) \leq cC^{-\beta}n^{-b}(\text{ Assumption 1 })$$

2. We have

$$\mathbb{P}(U_{a,n,s} > \mu_\star) = \mathbb{P}\left(\widehat{\mu}_{a,s} + C\frac{n^{\frac{b}{\beta}}}{s^{\frac{\alpha}{\beta}}} > \mu_\star\right) = \mathbb{P}\left(\widehat{\mu}_{a,s} - \mu_a > \Delta_a - C\frac{n^{\frac{b}{\beta}}}{s^{\frac{\alpha}{\beta}}}\right)$$

Because we choose

$$A_a(n) := \inf\left\{s \leq n : C\frac{n^{\frac{b}{\beta}}}{s^{\frac{\alpha}{\beta}}} \leq \frac{\Delta_a}{2}\right\} = \left(\frac{2C}{\Delta_a}\right)^{\frac{\beta}{\alpha}} n^{\frac{b}{\alpha}},$$

therefore,

$$\mathbb{P}(U_{a,n,s} > \mu_\star) \leq \mathbb{P}\left(\widehat{\mu}_{a,s} - \mu_a > C\frac{n^{\frac{b}{\beta}}}{s^{\frac{\alpha}{\beta}}}\right) \leq cC^{-\beta}n^{-b}(\text{ Assumption 1 })$$

that concludes the proof. $\qquad\square$

In turn, Lemma 6 permits us to prove the following crucial high-probability bound on the number of selection of each sub-optimal arm.

**Lemma 6.** *Consider a bandit problem defined as in Section 4.1. Assume $b > 1$. Let us define $A_a(n) := \inf\left\{s \leq n : C\frac{n^{\frac{b}{\beta}}}{s^{\frac{\alpha}{\beta}}} \leq \frac{\Delta_a}{2}\right\} = \left(\frac{2C}{\Delta_a}\right)^{\frac{\beta}{\alpha}} n^{\frac{b}{\alpha}}$. For all $u \geq A_a(n)$,*

$$\mathbb{P}\left(T_a(n) \geq u\right) \leq 2cC^{-\beta}\frac{(u-1)^{-(b-1)}}{b-1}.$$

*Proof.* For any $\tau \in \mathbb{R}$, we study two following events

$$\mathcal{E}_1 = \{\text{for each integer } t \in [u,n], \text{ we have } U_{a,t,u} \leq \tau\}, \tag{23}$$
$$\mathcal{E}_2 = \{\text{for each integer } t_0 \in [1, n-u], \text{ we have } U_{a_*,u+t_0,t_0} > \tau\}. \tag{24}$$

We want to prove that

$$\mathcal{E}_1 \cap \mathcal{E}_2 \Rightarrow T_a(n) \leq u.$$

Recall that

$$U_{a,t,s} = \widehat{\mu}_{a,s} + C\frac{t^{\frac{b}{\beta}}}{s^{\frac{\alpha}{\beta}}} \Rightarrow U_{a,t,u} = \widehat{\mu}_{a,u} + C\frac{t^{\frac{b}{\beta}}}{u^{\frac{\alpha}{\beta}}} \text{ and } U_{a_*,u+t_0,t_0} = \widehat{\mu}_{a_*,t_0} + C\frac{(u+t_0)^{\frac{b}{\beta}}}{t_0^{\frac{\alpha}{\beta}}}.$$

Then, for each $t_0$ such that $1 \leq t_0 \leq n - u$, and each $t$ such that $u + t_0 \leq t \leq n$,
We have

$$U_{a_*,t,t_0} = \widehat{\mu}_{a_*,t_0} + C\frac{t^{\frac{b}{\beta}}}{t_0^{\frac{\alpha}{\beta}}} \geq \widehat{\mu}_{a_*,t_0} + C\frac{(u+t_0)^{\frac{b}{\beta}}}{t_0^{\frac{\alpha}{\beta}}} > \tau > U_{a,t,u} = \widehat{\mu}_{a,u} + C\frac{t^{\frac{b}{\beta}}}{u^{\frac{\alpha}{\beta}}}. \tag{25}$$

We want to prove $T_a(n) \leq u$ by contradiction. Let assume that $T_a(n) > u$, then let denote $t'$ is the first time that the arm $a$ have been played $u$ times:

$$t' = \min\{t : t \leq n, T_a(n) = u\}.$$

Then at anytime t such that $t' < t \leq n$, meaning at any time $t$ after the arm $a$ has been selected $u$ time, from 25, we have

$$U_{a_*,t,t_0} > U_{a,t,u},$$

which mean the arm $a$ will not be selected after $u$ times, which contradicts our assumption that $T_a(n) > u$. Therefore

$$\mathcal{E}_1 \cap \mathcal{E}_2 \Rightarrow T_a(n) \leq u.$$

Then

$$\{T_a(n) \geq u\} \subset (\mathcal{E}_1^c \cup \mathcal{E}_2^c) = (\{\exists t : u \leq t \leq n, U_{a,t,u} > \tau\} \cup \{\exists t_0 : 1 \leq t_0 \leq n - u, U_{a_*,u+t_0,t_0} \leq \tau\}). \tag{26}$$

Therefore,

$$\mathbb{P}\left(T_a(n) \geq u\right) \leq \sum_{t=u}^{n}\mathbb{P}\left(U_{a,t,u} > \tau\right) + \sum_{t_0=1}^{n-u}\mathbb{P}\left(U_{a_*,u+t_0,t_0} \leq \tau\right). \tag{27}$$

We set $\tau = \mu_*$, and since $u \geq A_a(n)$, from Lemma 5, we have the following result

$$\sum_{t=u}^{n}\mathbb{P}\left(U_{a,t,u} > \tau\right) = \sum_{t=u}^{n}\mathbb{P}\left(U_{a,t,u} > \mu_*\right) \leq cC^{-\beta}\sum_{t=u}^{n}t^{-b} \leq cC^{-\beta}\int_{u-1}^{\infty}t^{-b}dt = cC^{-\beta}\frac{(u-1)^{-(b-1)}}{b-1} \tag{28}$$

Similarly,

$$\sum_{t_0=1}^{n-u}\mathbb{P}\left(U_{a_*,u+t_0,t_0} \leq \tau\right) = \sum_{t_0=1}^{n-u}\mathbb{P}\left(\widehat{\mu}_{a_*,t_0} + C\frac{(u+t_0)^{\frac{b}{\beta}}}{t_0^{\frac{\alpha}{\beta}}} > \mu_*\right) \leq cC^{-\beta}\sum_{t_0=1}^{n-u}(u+t_0)^{-b} \leq cC^{-\beta}\int_{u-1}^{\infty}t^{-b}dt \tag{29}$$

$$= cC^{-\beta}\frac{(u-1)^{-(b-1)}}{b-1}, \tag{30}$$

that concludes the proof. $\square$

**Lemma 7.** *Let us define the power mean estimator* $\widehat{\mu}_n(p)$ *as* $\widehat{\mu}_n(p) = \left( \sum_{a=1}^{K} \frac{T_a(n)}{n} \widehat{\mu}_{a,T_a(n)}^p \right)^{\frac{1}{p}}$. *For any* $p \geq 1$, *we have*

$$|\widehat{\mu}_n(p) - \mu_*| \leq R \sum_{a=1, a \neq a_*}^{K} \frac{T_a(n)}{n} + \left( \sum_{a=1}^{K} \frac{T_a(n)}{n} \left( |\widehat{\mu}_{a,T_a(n)} - \mu_a| \right)^p \right)^{\frac{1}{p}} \tag{31}$$

*Proof.* We observe that

$$\widehat{\mu}_{a,T_a(n)} \leq \mu_a + \left| \widehat{\mu}_{a,T_a(n)} - \mu_a \right|. \tag{32}$$

Since $\mu_* = \max_{a \in [K]}\{\mu_a\}$, we have

$$\widehat{\mu}_n(p) - \mu_* = \widehat{\mu}_n(p) - \sum_{a=1}^{K} T_a(n) \mu_* \leq \left( \sum_{a=1}^{K} \frac{T_a(n)}{n} \left( \widehat{\mu}_{a,T_a(n)} \right)^p \right)^{\frac{1}{p}} - \left( \sum_{a=1}^{K} \frac{T_a(n)}{n} (\mu_a)^p \right)^{\frac{1}{p}} \tag{33}$$

$$= \frac{\left( \sum_{a=1}^{K} T_a(n) \left( \widehat{\mu}_{a,T_a(n)} \right)^p \right)^{\frac{1}{p}} - \left( \sum_{a=1}^{K} T_a(n) (\mu_a)^p \right)^{\frac{1}{p}}}{n^{\frac{1}{p}}} \tag{34}$$

Applying Minkowski's inequality from Lemma 3(**Minkowski's inequality**), and the result of (32), we have

$$\widehat{\mu}_n(p) - \mu_* \leq \frac{\left( \sum_{a=1}^{K} T_a(n) \left( \mu_a + |\widehat{\mu}_{a,T_a(n)} - \mu_a| \right)^p \right)^{\frac{1}{p}} - \left( \sum_{a=1}^{K} T_a(n) (\mu_a)^p \right)^{\frac{1}{p}}}{n^{\frac{1}{p}}} \tag{35}$$

$$\leq \frac{\left( \sum_{a=1}^{K} T_a(n) \left( |\widehat{\mu}_{a,T_a(n)} - \mu_a| \right)^p \right)^{\frac{1}{p}}}{n^{\frac{1}{p}}} \tag{36}$$

On the other hand,

$$\mu_* - \widehat{\mu}_n(p) = \frac{n\mu_* - n\widehat{\mu}_n(p)}{n} = \frac{n\mu_* - \left( \sum_{a=1}^{K} T_a(n)\mu_a \right) + \sum_{a=1}^{K} T_a(n)\mu_a - n\widehat{\mu}_n(p)}{n} \tag{37}$$

$$= \frac{\sum_{a=1, a \neq a_*}^{K} T_a(n) |\mu_* - \mu_a| + \sum_{a=1}^{K} T_a(n)\mu_a - n\widehat{\mu}_n(p)}{n} \tag{38}$$

$$\leq R \sum_{a=1, a \neq a_*}^{K} \frac{T_a(n)}{n} + \sum_{a=1}^{K} \frac{T_a(n)}{n}\mu_a - \widehat{\mu}_n(p) \tag{39}$$

Because power mean is an increasing function of $p$, so that $\sum_{a=1}^{K} \frac{T_a(n)}{n}\mu_a \leq \left( \sum_{a=1}^{K} \frac{T_a(n)}{n} (\mu_a)^p \right)^{1/p}$. Furthermore, we observe that

$$\mu_a \leq \widehat{\mu}_{a,T_a(n)} + \left| \widehat{\mu}_{a,T_a(n)} - \mu_a \right|.$$

So that, from Equation (39) we have

$$\mu_* - \widehat{\mu}_n(p) \leq R \sum_{a=1, a \neq a_*}^{K} \frac{T_a(n)}{n} + \left( \sum_{a=1}^{K} \frac{T_a(n)}{n} (\mu_a)^p \right)^{1/p} - \widehat{\mu}_n(p)$$

$$\leq R \sum_{a=1, a \neq a_*}^{K} \frac{T_a(n)}{n} + \frac{\left( \sum_{a=1}^{K} T_a(n) \left( \widehat{\mu}_{a,T_a(n)} + |\widehat{\mu}_{a,T_a(n)} - \mu_a| \right)^p \right)^{\frac{1}{p}} - \left( \sum_{a=1}^{K} T_a(n) \left( \widehat{\mu}_{a,T_a(n)} \right)^p \right)^{\frac{1}{p}}}{n^{\frac{1}{p}}}$$

$$\leq R \sum_{a=1, a \neq a_*}^{K} \frac{T_a(n)}{n} + \frac{\left( \sum_{a=1}^{K} T_a(n) \left( |\widehat{\mu}_{a,T_a(n)} - \mu_a| \right)^p \right)^{\frac{1}{p}}}{n^{\frac{1}{p}}} \tag{40}$$

Therefore, from equation (36), and equation (40), we can derive

$$|\widehat{\mu}_n(p) - \mu_*| \leq R \sum_{a=1, a\neq a_*}^{K} \frac{T_a(n)}{n} + \left( \sum_{a=1}^{K} \frac{T_a(n)}{n} \left( |\widehat{\mu}_{a, T_a(n)} - \mu_a| \right)^p \right)^{\frac{1}{p}},$$

that concludes the proof. $\qquad\square$

**Lemma 8.** *Consider a bandit problem defined as in Section 4.1. With $R \geq \epsilon \geq n^{-\frac{\alpha}{\beta}}$, we have*

$$\mathbb{P}\left( \frac{T_{a_*}(n)}{n} \left( |\widehat{\mu}_{a_*, T_{a_*}(n)} - \mu_*| \right)^p > \epsilon^p \right) \leq \frac{2cC^{-\beta}(K-1)A(n)^{-(b-1)}}{b-1} + \frac{c}{\alpha - 1}\epsilon^{-\beta}(n - (K-1)(A(n)+1) - 1)^{-\alpha+1}$$

(41)

*Proof.* We have

$$\mathbb{P}\left( \frac{T_{a_*}(n)}{n} \left( |\widehat{\mu}_{a_*, T_{a_*}(n)} - \mu_*| \right)^p > \epsilon^p \right) \leq \mathbb{P}\left( |\widehat{\mu}_{a_*, T_{a_*}(n)} - \mu_*| > \epsilon \right).$$

Applying results of Lemma 4, we have

$$\mathbb{P}\left( |\widehat{\mu}_{a_\star, T_{a_*}(n)} - \mu_\star| > \epsilon \right) \leq \underbrace{\sum_{a \neq a_*}^{K} \mathbb{P}\left( T_a(n) > A(n) + 1 \right)}_{F_{11}} + \underbrace{\frac{c}{\alpha - 1}\epsilon^{-\beta}(n - (K-1)(A(n)+1) - 1)^{-\alpha+1}}_{F_{12}}. \qquad (42)$$

From the result of Lemma 6, with $b > 1$, we also have

$$F_{11} \leq \sum_{a=1, a\neq a_*}^{K} \mathbb{P}\left( T_a(n) > A(n) + 1 \right) \leq \sum_{a=1, a\neq a_*}^{K} 2cC^{-\beta} \frac{A(n)^{-(b-1)}}{b-1} = \frac{2cC^{-\beta}(K-1)A(n)^{-(b-1)}}{b-1}$$

that concludes the proof. $\qquad\square$

**Lemma 9.** *Consider a bandit problem defined as in Section 4.1. With a is a suboptimal arm, $R \geq \epsilon \geq n^{-\frac{\alpha}{\beta}}$, we can find a constant $N_0$ such that $\forall n \geq N_0$, such that*

- *With $1 \leq p \leq 2, \alpha \leq \frac{\beta}{p}$, we have*

$$\mathbb{P}\left( \frac{T_a(n)}{n} \left( |\widehat{\mu}_{a, T_a(n)} - \mu_a| \right)^p > \frac{1}{K-1}\epsilon^p \right) \leq \frac{2cC^{-\beta}}{(b-1)}A(n)^{-(b-1)} + \frac{2c(K-1)^{\frac{\beta}{p}}}{-(\alpha - \frac{\beta}{p}) + 1}\epsilon^{-\beta}(A_a(n) + 1)^{-(\alpha-1)}.$$

(43)

- *With $p > 2$, and $0 < \alpha - \frac{\beta}{p} < 1$, we have*

$$\mathbb{P}\left( \frac{T_a(n)}{n} \left( |\widehat{\mu}_{a, T_a(n)} - \mu_a| \right)^p > \frac{1}{K-1}\epsilon^p \right) \leq \frac{2cC^{-\beta}}{(b-1)}A(n)^{-(b-1)} + \frac{c(K-1)^{\frac{\beta}{p}}}{1 - (\alpha - \frac{\beta}{p})}\epsilon^{-\beta}(A(n) + 1)^{-(\alpha-1)}. \quad (44)$$

- *With $p > 2$, and $\alpha - \frac{\beta}{p} > 1$, we have*

$$\mathbb{P}\left( \frac{T_a(n)}{n} \left( |\widehat{\mu}_{a, T_a(n)} - \mu_a| \right)^p > \frac{1}{K-1}\epsilon^p \right) \leq \frac{2cC^{-\beta}}{(b-1)}A(n)^{-(b-1)} + \frac{c(K-1)^{\frac{\beta}{p}}(\alpha - \frac{\beta}{p})}{(\alpha - \frac{\beta}{p}) - 1}\epsilon^{-\beta}(A(n) + 1)^{-\frac{\beta}{p}} \quad (45)$$

*Proof.* Recall that $\forall u > A_a(n) = \left( \frac{2Cn^{\frac{b}{\beta}}}{\triangle_a} \right)^{\frac{\beta}{\alpha}}$,

$$\mathbb{P}\left( T_a(n) > u \right) \leq 2cC^{-\beta} \frac{(u-1)^{-(b-1)}}{b-1}.$$

We consider 2 events, $\mathcal{E}_1 = \{T_a(n) > A_a(n) + 1)\}$, and $\mathcal{E}_1^c = \{T_a(n) \leq A_a(n) + 1\}$, then

$$\mathbb{P}\left(\frac{T_a(n)}{n}\left(|\widehat{\mu}_{a,T_a(n)} - \mu_a|\right)^p > \frac{1}{K-1}\epsilon^p\right) \leq \mathbb{P}\left(T_a(n) > A_a(n) + 1\right)$$

$$+ \mathbb{P}\left(T_a(n) \leq A_a(n) + 1; \frac{T_a(n)}{n}\left(|\widehat{\mu}_{a,T_a(n)} - \mu_a|\right)^p > \frac{1}{K-1}\epsilon^p\right)$$

$$\leq \underbrace{2cC^{-\beta}\frac{A_a(n)^{-(b-1)}}{b-1}}_{G_1} + \underbrace{\sum_{t=1}^{A_a(n)+1}\mathbb{P}\left(\frac{t}{n}|\widehat{\mu}_{a,t} - \mu_a|^p > \frac{1}{K-1}\epsilon^p\right)}_{G2}$$

For $G_2$, we can see that we can find $N_0$ such that with $t \leq A(n)+1, \forall n \geq N_0$, $\left(\frac{n}{t(K-1)}\right)^{\frac{1}{p}}\epsilon > \epsilon \geq n^{-\frac{\alpha}{\beta}}$. Therefore,

$$G_2 \leq \sum_{t=1}^{A_a(n)+1}\mathbb{P}\left(|\widehat{\mu}_{a,t} - \mu_a| > \left(\frac{n}{t(K-1)}\right)^{\frac{1}{p}}\epsilon\right) \leq \sum_{t=1}^{A_a(n)+1} ct^{-\alpha}\left(\left(\frac{n}{t(K-1)}\right)^{\frac{1}{p}}\epsilon\right)^{-\beta} \leq \sum_{t=1}^{A_a(n)+1} c(K-1)^{\frac{\beta}{p}}t^{-(\alpha-\frac{\beta}{p})}\epsilon^{-\beta}n^{-\frac{\beta}{p}}.$$

We study 2 cases:

Case 1: $\alpha - \frac{\beta}{p} \leq 0$, which can only happen if $p \leq 2$ because $\alpha \leq \frac{\beta}{2}$, and actually when $\alpha \leq \frac{\beta}{2}$ we just need $1 \leq p \leq 2$, then

$$G_2 \leq c(K-1)^{\frac{\beta}{p}}\epsilon^{-\beta}n^{-\frac{\beta}{p}}\left(\int_1^{A_a(n)+1} t^{-(\alpha-\frac{\beta}{p})}dt + (A_a(n)+1)^{-(\alpha-\frac{\beta}{p})}\right)$$

$$= c(K-1)^{\frac{\beta}{p}}\epsilon^{-\beta}n^{-\frac{\beta}{p}}\left(\left(\frac{t^{-(\alpha-\frac{\beta}{p})+1}}{-(\alpha-\frac{\beta}{p})+1}+C\right)\Big|_1^{A_a(n)+1} + (A_a(n)+1)^{-(\alpha-\frac{\beta}{p})}\right)$$

$$\leq c(K-1)^{\frac{\beta}{p}}\epsilon^{-\beta}(A_a(n)+1)^{-\frac{\beta}{p}}\left(\frac{(A_a(n)+1)^{-(\alpha-\frac{\beta}{p})+1}}{-(\alpha-\frac{\beta}{p})+1} - \frac{1}{-(\alpha-\frac{\beta}{p})+1} + (A_a(n)+1)^{-(\alpha-\frac{\beta}{p})}\right).$$

Because $-(\alpha-\frac{\beta}{p})+1 \geq 1$, we can find a constant $N_\epsilon$ such that $\forall n \geq N_\epsilon$, we have

$$G_2 \leq 2c(K-1)^{\frac{\beta}{p}}\epsilon^{-\beta}(A_a(n)+1)^{-\frac{\beta}{p}}\frac{(A_a(n)+1)^{-(\alpha-\frac{\beta}{p})+1}}{-(\alpha-\frac{\beta}{p})+1} = \frac{2c(K-1)^{\frac{\beta}{p}}}{-(\alpha-\frac{\beta}{p})+1}\epsilon^{-\beta}(A_a(n)+1)^{-(\alpha-1)}.$$

Therefore, we have

$$\mathbb{P}\left(\frac{T_a(n)}{n}\left(|\widehat{\mu}_{a,T_a(n)} - \mu_a|\right)^p > \frac{1}{K}\epsilon^p\right) \leq \frac{2cC^{-\beta}}{(b-1)}A(n)^{-(b-1)} + \frac{2c(K-1)^{\frac{\beta}{p}}}{-(\alpha-\frac{\beta}{p})+1}\epsilon^{-\beta}(A_a(n)+1)^{-(\alpha-1)}. \qquad (46)$$

that concludes for the Inequality 43.

Case 2: $\alpha - \frac{\beta}{p} > 0$, which can only happen if $p > 2$ because $\alpha \leq \frac{\beta}{2}$. We have

$$\sum_{t=1}^{A_a(n)+1} t^{-(\alpha-\frac{\beta}{p})} \leq 1 + \int_1^{A_a(n)+1} t^{-(\alpha-\frac{\beta}{p})}dt = 1 + \left(\frac{t^{-(\alpha-\frac{\beta}{p})+1}}{-(\alpha-\frac{\beta}{p})+1}+C\right)\Big|_1^{A_a(n)+1}$$

$$= 1 + \frac{(A_a(n)+1)^{-(\alpha-\frac{\beta}{p})+1}}{-(\alpha-\frac{\beta}{p})+1} - \frac{1}{-(\alpha-\frac{\beta}{p})+1}$$

$$= \frac{\alpha - \frac{\beta}{p}}{\alpha - \frac{\beta}{p} - 1} - \frac{(A_a(n)+1)^{-(\alpha-\frac{\beta}{p})+1}}{(\alpha - \frac{\beta}{p}) - 1},$$

so that

$$G_2 \leq c(K-1)^{\frac{\beta}{p}}\left(\frac{\alpha - \frac{\beta}{p}}{\alpha - \frac{\beta}{p} - 1} - \frac{(A_a(n)+1)^{-(\alpha-\frac{\beta}{p})+1}}{(\alpha - \frac{\beta}{p}) - 1}\right)\epsilon^{-\beta}n^{-\frac{\beta}{p}}$$

$$= c(K-1)^{\frac{\beta}{p}}\left(\frac{(A_a(n)+1)^{-(\alpha-\frac{\beta}{p})+1}}{1 - (\alpha - \frac{\beta}{p})} - \frac{\alpha - \frac{\beta}{p}}{1 - (\alpha - \frac{\beta}{p})}\right)\epsilon^{-\beta}(A(n)+1)^{-\frac{\beta}{p}}.$$

If $0 < \alpha - \frac{\beta}{p} < 1$, then we can find a constant $N_{G2}$ such that $\forall n \geq N_{G2}$, we have

$$G_2 \leq \frac{c(K-1)^{\frac{\beta}{p}}}{1 - (\alpha - \frac{\beta}{p})} \epsilon^{-\beta}(A(n)+1)^{-(\alpha-1)} = \frac{c(K-1)^{\frac{\beta}{p}}}{1 - (\alpha - \frac{\beta}{p})} \epsilon^{-\beta}(A(n)+1)^{-(\alpha-1)}.$$

Therefore,

$$\mathbb{P}\left(\frac{T_a(n)}{n}\left(|\widehat{\mu}_{a,T_a(n)} - \mu_a|\right)^p > \frac{1}{K}\epsilon^p\right) \leq \frac{2cC^{-\beta}}{(b-1)}A(n)^{-(b-1)} + \frac{c(K-1)^{\frac{\beta}{p}}}{1 - (\alpha - \frac{\beta}{p})}\epsilon^{-\beta}(A(n)+1)^{-(\alpha-1)}. \qquad (47)$$

that concludes for the Inequality 44.

If $\alpha - \frac{\beta}{p} > 1$, we can find a constant $N_0$ such that $\forall n \geq N_0$, we have

$$G_2 \leq c(K-1)^{\frac{\beta}{p}}\left(\frac{\alpha - \frac{\beta}{p}}{\alpha - \frac{\beta}{p} - 1} - \frac{(A(n)+1)^{-(\alpha-\frac{\beta}{p})+1}}{(\alpha - \frac{\beta}{p}) - 1}\right)\epsilon^{-\beta}(A(n)+1)^{-\frac{\beta}{p}} \leq \frac{c(K-1)^{\frac{\beta}{p}}(\alpha - \frac{\beta}{p})}{(\alpha - \frac{\beta}{p}) - 1}\epsilon^{-\beta}(A(n)+1)^{-\frac{\beta}{p}},$$

that concludes for the Inequality 45

$$\mathbb{P}\left(\frac{T_a(n)}{n}\left(|\widehat{\mu}_{a,T_a(n)} - \mu_a|\right)^p > \frac{1}{K}\epsilon^p\right) \leq \frac{2cC^{-\beta}}{(b-1)}A(n)^{-(b-1)} + \frac{c(K-1)^{\frac{\beta}{p}}(\alpha - \frac{\beta}{p})}{(\alpha - \frac{\beta}{p}) - 1}\epsilon^{-\beta}(A(n)+1)^{-\frac{\beta}{p}}. \qquad (48)$$

Actually we should not choose our parameters to fall in this case because when p is big $-\frac{\beta}{p}$ will get big and the bound is looser. $\qquad \square$

**Lemma 10.** *Consider a bandit problem defined as in Section 4.1. Let us define the power mean estimator $\widehat{\mu}_n(p)$ as*
$\widehat{\mu}_n(p) = \left(\sum_{a=1}^{K}\frac{T_a(n)}{n}\widehat{\mu}_{a,T_a(n)}^p\right)^{\frac{1}{p}}$. *Define $A(n) = \left(\frac{2Cn^{\frac{b}{\beta}}}{\triangle}\right)^{\frac{\beta}{\alpha}}$, where $\triangle = \min_{a \in [K]}\{\triangle_a\}, \triangle_a = \mu_* - \mu_a$. Let $\epsilon_0 = $*
$\frac{2^{\frac{1}{p}}n\epsilon}{x} + \frac{nR(K-1)}{x}\left(\frac{2^{\frac{1}{p}}(3+A(n))x}{n}\right), R \geq \epsilon \geq n^{-\frac{\alpha}{\beta}}$. *We can find a constant $N_p$ such that for any $n \geq N_p$ and $x \geq 1$, such that*

$$\mathbb{P}\left(|\widehat{\mu}_n(p) - \mu_*| \geq \frac{\epsilon_0 x}{n}\right) \leq \frac{8cC^{-\beta}KR^{\beta}\epsilon^{-\beta}A(n)^{-(b-1)}}{b-1} + 2cC^{-\beta}(K-1)\frac{(2^{\frac{1}{p}}(3+A(n))x - 1)^{-(b-1)}}{b-1}. \qquad (49)$$

*Proof.* As the results from Lemma 7, we can derive

$$|\widehat{\mu}_n(p) - \mu_*| \leq R\sum_{a=1,a\neq a_*}^{K}\frac{T_a(n)}{n} + \left(\sum_{a=1}^{K}\frac{T_a(n)}{n}\left(|\widehat{\mu}_{a,T_a(n)} - \mu_a|\right)^p\right)^{\frac{1}{p}} \qquad (50)$$

Because $\frac{\epsilon_0 x}{n} = 2^{\frac{1}{p}}\epsilon + R(K-1)\left(\frac{2^{\frac{1}{p}}(3+A(n))x}{n}\right)$, so that

$$\Rightarrow \mathbb{P}\left(|\widehat{\mu}_n(p) - \mu_a| > \frac{\epsilon_0 x}{n}\right) \leq \underbrace{\mathbb{P}\left(R\sum_{a=1,a\neq a_*}^{K}\frac{T_a(n)}{n} > R(K-1)(\frac{2^{\frac{1}{p}}(3+A(n))x}{n})\right)}_{H_1}$$

$$+ \underbrace{\mathbb{P}\left(\left(\sum_{a=1}^{K}\frac{T_a(n)}{n}\left(|\widehat{\mu}_{a,T_a(n)} - \mu_a|\right)^p\right)^{\frac{1}{p}} \geq 2^{\frac{1}{p}}\epsilon\right)}_{H_2}$$

To upper bound $H_1$: with $x \geq 1$ We have

$$H_1 \leq \sum_{a=1, a \neq a_*}^{K} \mathbb{P}\left(\frac{T_a(n)}{n} > \frac{2^{\frac{1}{p}}(3 + A(n))x}{n}\right) = \sum_{a=1, a \neq a_*}^{K} \mathbb{P}\left(T_a(n) > 2^{\frac{1}{p}}(3 + A(n))x\right)$$

$$\overset{\text{(Lemma 6)}}{\leq} 2cC^{-\beta}(K-1)\frac{(2^{\frac{1}{p}}(3 + A(n))x - 1)^{-(b-1)}}{b-1} \tag{51}$$

To upper bound $H_2$:

$$H_2 = \mathbb{P}\left(\sum_{a=1}^{K} \frac{T_a(n)}{n}\left(|\widehat{\mu}_{a,T_a(n)} - \mu_a|\right)^p > 2\epsilon^p\right)$$

$$\leq \underbrace{\mathbb{P}\left(\frac{T_{a_*}(n)}{n}\left(|\widehat{\mu}_{a_*,T_{a_*}(n)} - \mu_{a_*}|\right)^p > \epsilon^p\right)}_{F_1} + \underbrace{\sum_{a=1, a \neq a_*}^{K} \mathbb{P}\left(\frac{T_a(n)}{n}\left(|\widehat{\mu}_{a,T_a(n)} - \mu_a|\right)^p > \frac{1}{K-1}\epsilon^p\right)}_{F_2}$$

With $F_1$: According to Lemma 8, we can find a constant $N_0$, such that $\forall n \geq N_0$, we have

$$F_1 \leq \frac{2cC^{-\beta}(K-1)A(n)^{-(b-1)}}{b-1} + \frac{c}{\alpha-1}\epsilon^{-\beta}(n - (K-1)(A(n)+1) - 1)^{-\alpha+1} \tag{52}$$

With $F_2$: According to Lemma 9, we can find a constant $N_0$, such that $\forall n \geq N_0$, we have

- With $1 \leq p \leq 2, \alpha \leq \frac{\beta}{p}$, we have

$$F_2 \leq \frac{2cC^{-\beta}}{(b-1)}A(n)^{-(b-1)} + \frac{2c(K-1)^{\frac{\beta}{p}}}{-(\alpha - \frac{\beta}{p}) + 1}\epsilon^{-\beta}(A_a(n) + 1)^{-(\alpha-1)}. \tag{53}$$

- With $p > 2$, and $0 < \alpha - \frac{\beta}{p} < 1$, we have

$$F_2 \leq \frac{2cC^{-\beta}}{(b-1)}A(n)^{-(b-1)} + \frac{c(K-1)^{\frac{\beta}{p}}}{-(\alpha - \frac{\beta}{p}) + 1}\epsilon^{-\beta}(A_a(n) + 1)^{-(\alpha-1)}. \tag{54}$$

So that

$$H_2 \leq F_1 + F_2 \leq \frac{2cC^{-\beta}(K-1)A(n)^{-(b-1)}}{b-1} + \frac{c}{\alpha-1}\epsilon^{-\beta}(n - (K-1)(A(n)+1) - 1)^{-\alpha+1} \tag{55}$$

$$+ \frac{2cC^{-\beta}}{(b-1)}A(n)^{-(b-1)} + \frac{c(K-1)^{\frac{\beta}{p}}}{-(\alpha - \frac{\beta}{p}) + 1}\epsilon^{-\beta}(A_a(n) + 1)^{-(\alpha-1)}$$

where $1 \leq p \leq 2, \alpha \leq \frac{\beta}{p}$ or $p > 2$, and $0 < \alpha - \frac{\beta}{p} < 1$.

Because $b - 1 < \alpha - 1, n^{-\frac{\alpha}{\beta}} \leq \epsilon \leq R$, so that we can find a constant $N_p$ such that $\forall n \geq N_p$

$$H_2 \leq \frac{8cC^{-\beta}K\left(\frac{R}{\epsilon}\right)^{\beta}A(n)^{-(b-1)}}{b-1} = \frac{8cC^{-\beta}KR^{\beta}\epsilon^{-\beta}A(n)^{-(b-1)}}{b-1}. \tag{56}$$

Combining 51 and 56, we can conclude the proof. $\qquad\square$

**Theorem 1.** *For $a \in [K]$, let $(\widehat{\mu}_{a,n})_{n \geq 1}$ be a sequence of estimators satisfying $\widehat{\mu}_{a,n} \xrightarrow[n \to \infty]{\alpha, \beta} \mu_a$ and let $\mu_\star = \max_a\{\mu_a\}$. Assume that the arms are sampled according to the strategy equation 5 with parameters $\alpha, \beta, b$ and $C$. Assume that $p, \alpha, \beta$ and $b$ satisfy one of these two conditions:*

*(i) $1 \leq p \leq 2$ and $\alpha \leq \frac{\beta}{2}$*

*(ii) $p > 2$ and $0 < \alpha - \frac{\beta}{p} < 1$*

If $\alpha \left(1 - \frac{b}{\alpha}\right) \leq b < \alpha$ then the sequence of estimators $\widehat{\mu}_n(p)$ satisfies

$$\widehat{\mu}_n(p) \xrightarrow[n \to \infty]{\alpha', \beta'} \mu_\star$$

for $\alpha' = (b-1)\left(1 - \frac{b}{\alpha}\right)$ and $\beta' = (b-1)$ *for some value of the constant $C$ in equation 5 that depends on $K, b, \alpha, p, \Delta_{\min}$ with $\Delta_{\min} = \min_{a:\mu_a < \mu_\star}(\mu_\star - \mu_a)$.*

*Proof.* We will use the results of Lemma 10 to derive the proof of Theorem 1. We want to have an upper bound

$$D = \mathbb{P}\left( |\widehat{\mu}_n(p) - \mu_*| \geq \epsilon \right).$$

Due to Lemma 10, we have

$$\epsilon_0 = \frac{2^{\frac{1}{p}} n \epsilon'}{x} + \frac{nR(K-1)}{x}(\frac{2^{\frac{1}{p}}(3 + A(n))x}{n}) \Rightarrow \frac{\epsilon_0 x}{n} = 2^{\frac{1}{p}}\epsilon' + R(K-1)\left(\frac{2^{\frac{1}{p}}(3 + A(n))x}{n}\right).$$

Also, from Lemma 10, recall that $A(n) = \left(\frac{2Cn^{\frac{b}{\beta}}}{\triangle}\right)^{\frac{\beta}{\alpha}}$, we study

$$\epsilon = 2^{\frac{1}{p}} R(2K-1)\left(\frac{\left(\frac{2C}{\triangle}\right)^{\frac{\beta}{\alpha}} n^{\frac{b}{\beta}} x}{n}\right) = 2^{\frac{1}{p}} R(2K-1)\left(\frac{A(n)x}{n}\right). \tag{57}$$

We want to find $N_0 > 0$, that for any $n \geq N_0, \epsilon \geq \frac{\epsilon_0 x}{n}$. To do that, we compute

$$\epsilon - \frac{\epsilon_0 x}{n} = 2^{\frac{1}{p}} R(2K-1)\left(\frac{A(n)x}{n}\right) - R(K-1)\left(\frac{2^{\frac{1}{p}}(3 + A(n))x}{n}\right) - 2^{\frac{1}{p}}\epsilon'$$

$$= 2^{\frac{1}{p}} R(2K-1)\left(\frac{A(n)x}{n}\right) - 2^{\frac{1}{p}} R(K-1)(\frac{A(n)x}{n}) - 2^{\frac{1}{p}} R(K-1)(\frac{3x}{n}) - 2^{\frac{1}{p}}\epsilon'$$

$$= 2^{\frac{1}{p}} RK(\frac{A(n)x}{n}) - 2^{\frac{1}{p}} R(K-1)(\frac{3x}{n}) - 2^{\frac{1}{p}}\epsilon' = \underbrace{2^{\frac{1}{p}} R(K-1)(\frac{x}{n})(A(n) - 3)}_{T_1} + \underbrace{2^{\frac{1}{p}} R(\frac{A(n)x}{n}) - 2^{\frac{1}{p}}\epsilon'}_{T_2}$$

Because $A(n) \sim \Theta(n^{\frac{b}{\alpha}})$ and $\frac{b}{\alpha} > 0$, then $\exists N_1 > 0$ with $n \geq N_1$ that $T_1 > 0$. We can see that $\frac{A(n)}{n} \sim \Theta(n^{-(1-\frac{b}{\alpha})})$. We choose $\epsilon' = (n^{-\frac{\alpha}{\beta}}x)$ that satisfies condition $\epsilon' \geq n^{-\frac{\alpha}{\beta}}$. With $c \geq 1$, and $\frac{R}{\triangle^{\frac{\beta}{\alpha}}} > 1$, We have

$$\frac{RA(n)x}{n} = \frac{R\left(\frac{2Cn^{\frac{b}{\beta}}}{\triangle}\right)^{\frac{\beta}{\alpha}} x}{n} = (2C)\frac{R}{\triangle^{\frac{\beta}{\alpha}}} n^{-(1-\frac{b}{\alpha})}x \geq n^{-(1-\frac{b}{\alpha})}x$$

and because $1 - \frac{b}{\alpha} \leq \frac{1}{2} \leq \frac{\alpha}{\beta}$ then

$$T_2 = 2^{\frac{1}{p}}\left(R(\frac{A(n)x}{n}) - n^{-\frac{\alpha}{\beta}}x\right) \geq 0.$$

Then we can define $N_0 = \min\{t : R(2K-1)\left(\frac{A(t)x}{t}\right) - R(K-1)\left(\frac{(3+A(t))x}{t}\right) - \epsilon' \geq 0\}$, therefore with $n \geq N_0$, According to Lemma 10, with $1 \leq p \leq 2, \alpha \leq \frac{\beta}{p}$ or $p > 2; 0 < \alpha - \frac{\beta}{p} < 1$ , we have

$$D = \mathbb{P}\left( |\widehat{\mu}_n(p) - \mu_*| \geq \epsilon \right) \leq \mathbb{P}\left( |\widehat{\mu}_n(p) - \mu_*| \geq \frac{\epsilon_0 x}{n} \right)$$

$$\leq \frac{8cC^{-\beta} K R^\beta \epsilon^{-\beta} A(n)^{-(b-1)}}{b-1} + 2cC^{-\beta}(K-1)\frac{(2^{\frac{1}{p}}(3 + A(n))x - 1)^{-(b-1)}}{b-1} \quad \text{(Lemma 10)}$$

Furthermore, we observe that $2^{\frac{1}{p}}(3 + A(n))x - 1 > A(n)x$ with $x \geq 1$. So that,

$$
\begin{aligned}
D &\leq \frac{8cC^{-\beta}KR^{\beta}(n^{-(1-\frac{b}{\alpha})}x)^{-\beta}A(n)^{-(b-1)}}{b-1} + 2cC^{-\beta}(K-1)\frac{(A(n)x)^{-(b-1)}}{b-1} \\
&\leq \frac{8cC^{-\beta}KR^{\beta}n^{\beta(1-\frac{b}{\alpha})}A(n)^{-(b-1)}x^{-\beta}}{b-1} + 2cC^{-\beta}(K-1)\frac{A(n)^{-(b-1)}x^{-(b-1)}}{b-1} \\
&= \frac{8cC^{-\beta}KR^{\beta}n^{\beta(1-\frac{b}{\alpha})}\left(\left(\frac{2Cn^{\frac{b}{\beta}}}{\triangle}\right)^{\frac{\beta}{\alpha}}\right)^{-(b-1)}x^{-\beta}}{b-1} + 2cC^{-\beta}(K-1)\frac{\left(\left(\frac{2Cn^{\frac{b}{\beta}}}{\triangle}\right)^{\frac{\beta}{\alpha}}\right)^{-(b-1)}x^{-(b-1)}}{b-1} \\
&= \frac{8cC^{-\beta}KR^{\beta}n^{\beta(1-\frac{b}{\alpha})}n^{-\frac{b}{\alpha}(b-1)}\left(\frac{2C}{\triangle}\right)^{-\frac{\beta}{\alpha}(b-1)}x^{-\beta}}{b-1} + 2cC^{-\beta}(K-1)\frac{\left(\frac{2C}{\triangle}\right)^{-\frac{\beta}{\alpha}(b-1)}n^{-\frac{b}{\alpha}(b-1)}x^{-(b-1)}}{b-1} \\
&= \frac{8cC^{-\beta}KR^{\beta}\left(\frac{2C}{\triangle}\right)^{-\frac{\beta}{\alpha}(b-1)}n^{\beta(1-\frac{b}{\alpha})}n^{-\frac{b}{\alpha}(b-1)}x^{-\beta}}{b-1} + 2cC^{-\beta}(K-1)\frac{\left(\frac{2C}{\triangle}\right)^{-\frac{\beta}{\alpha}(b-1)}n^{-\frac{b}{\alpha}(b-1)}x^{-(b-1)}}{b-1}
\end{aligned}
$$

From (57), we have $A(n)x = \frac{n\epsilon}{(2^{\frac{1}{p}}R(2K-1))}$, and $x = \frac{1}{(2^{\frac{1}{p}}R(2K-1))}\epsilon n^{-(\frac{b}{\alpha}-1)}\left(\frac{2C}{\triangle}\right)^{\frac{-\beta}{\alpha}}$. Therefore,

$$
\begin{aligned}
D &\leq \frac{8cC^{-\beta}KR^{\beta}\left(\frac{2C}{\triangle}\right)^{-\frac{\beta}{\alpha}(b-1)}n^{\beta(1-\frac{b}{\alpha})}n^{-\frac{b}{\alpha}(b-1)}}{b-1}\left(\frac{1}{(2^{\frac{1}{p}}R(2K-1))}\epsilon n^{-(\frac{b}{\alpha}-1)}\left(\frac{2C}{\triangle}\right)^{\frac{-\beta}{\alpha}}\right)^{-\beta} \\
&\quad + 2cC^{-\beta}(K-1)\frac{\left(\frac{2C}{\triangle}\right)^{-\frac{\beta}{\alpha}(b-1)}n^{-\frac{b}{\alpha}(b-1)}}{b-1}\left(\frac{1}{(2^{\frac{1}{p}}R(2K-1))}\epsilon n^{-(\frac{b}{\alpha}-1)}\left(\frac{2C}{\triangle}\right)^{\frac{-\beta}{\alpha}}\right)^{-(b-1)} \\
&\leq \frac{8cC^{-\beta}KR^{\beta}\left(\frac{2C}{\triangle}\right)^{-\frac{\beta}{\alpha}(b-1)}}{b-1}\left(\frac{1}{(2^{\frac{1}{p}}R(2K-1))}\left(\frac{2C}{\triangle}\right)^{\frac{-\beta}{\alpha}}\right)^{-\beta}\epsilon^{-\beta}n^{-\beta(1-\frac{b}{\alpha})}n^{\beta(1-\frac{b}{\alpha})}n^{-\frac{b}{\alpha}(b-1)} \\
&\quad + 2cC^{-\beta}(K-1)\frac{\left(\frac{2C}{\triangle}\right)^{-\frac{\beta}{\alpha}(b-1)}}{b-1}\left(\frac{1}{(2^{\frac{1}{p}}R(2K-1))}\left(\frac{2C}{\triangle}\right)^{\frac{-\beta}{\alpha}}\right)^{-(b-1)}\epsilon^{-(b-1)}n^{-(b-1)(1-\frac{b}{\alpha})}n^{-\frac{b}{\alpha}(b-1)} \\
&\leq c_0 n^{-\alpha'}\left(\frac{\epsilon}{R}\right)^{-\beta'} = c_0 R^{\beta'}n^{-\alpha'}\epsilon^{-\beta'}
\end{aligned}
$$

with

$$
c_0 = 2\max\left\{\frac{8cC^{-\beta}K\left(\frac{2C}{\triangle}\right)^{-\frac{\beta}{\alpha}(b-1)}}{b-1}\left(\frac{\left(\frac{2C}{\triangle}\right)^{\frac{-\beta}{\alpha}}}{(2^{\frac{1}{p}}R(2K-1))}\right)^{-\beta}, \frac{2cC^{-\beta}(K-1)\left(\frac{2C}{\triangle}\right)^{-\frac{\beta}{\alpha}(b-1)}}{b-1}\left(\frac{\left(\frac{2C}{\triangle}\right)^{\frac{-\beta}{\alpha}}}{(2^{\frac{1}{p}}(2K-1))}\right)^{-(b-1)}\right\}
$$

(58)

$$
= \frac{16cC^{-\beta}K\left(\frac{2C}{\triangle}\right)^{-\frac{\beta}{\alpha}(b-1)}}{b-1}\left(\frac{\left(\frac{2C}{\triangle}\right)^{\frac{-\beta}{\alpha}}}{(2^{\frac{1}{p}}R(2K-1))}\right)^{-\beta} \quad \text{because} \quad \left(\frac{\left(\frac{2C}{\triangle}\right)^{\frac{-\beta}{\alpha}}}{(2^{\frac{1}{p}}R(2K-1))}\right) < 1 \text{ and } 2K > 2(K-1)
$$

$$
\alpha' = \min\{\frac{b}{\alpha}(b-1), b-1\} = \frac{b}{\alpha}(b-1)
$$

$$
\beta' = \max\{b-1, \beta\} = b-1
$$

But we need $\epsilon \geq n^{-\frac{\alpha'}{\beta'}}$. Then with the condition $1 - \frac{b}{\alpha} \leq \frac{b}{\alpha} \Rightarrow \alpha(1-\frac{b}{\alpha}) \leq b$, we can choose

$$
\alpha' = (b-1)(1-\frac{b}{\alpha}),
$$

$$
\beta' = (b-1),
$$

and according to (58)

$$
c' = c_0 R^{\beta} = \frac{8cC^{-\beta}KR^{\beta}\left(\frac{2C}{\triangle}\right)^{-\frac{\beta}{\alpha}(b-1)}}{b-1}\left(\frac{\left(\frac{2C}{\triangle}\right)^{\frac{-\beta}{\alpha}}}{(2^{\frac{1}{p}}R(2K-1))}\right)^{-\beta} = \frac{2^{b+\frac{\beta}{p}}cC^{-\beta}K(2K-1)^{\beta}R^{2\beta}}{(b-1)}\left(\frac{2C}{\triangle}\right)^{-\frac{\beta}{\alpha}(b-1-\beta)}.
$$

This inequality is only correct for $n \geq N_0$. We want the inequality to be correct for all $n$. We want to show that the following inequality is correct for all $N_0 > n \geq 1$

$$\mathbb{P}\left(|\widehat{\mu}_n(p) - \mu_*| \geq \epsilon\right) \leq c' n^{-(b-1)(1-\frac{b}{\alpha})} \epsilon^{-(b-1)}.$$

We have $|\widehat{\mu}_n(p) - \mu_*| \leq R$. We choose $\epsilon$ as the form $RN_0\epsilon$. Then we have to prove that for $1 \leq n < N_0$,

$$D = \mathbb{P}\left(|\widehat{\mu}_n(p) - \mu_*| \geq RN_0\epsilon\right) \leq c' n^{-(b-1)(1-\frac{b}{\alpha})}(R\epsilon N_0)^{-(b-1)} = c'\left(\frac{1}{RN_0}\right)^{(b-1)}\left(\frac{n^{-(1-\frac{b}{\alpha})}}{\epsilon}\right)^{(b-1)}.$$

$$= \underbrace{c'\left(\frac{1}{RN_0}\right)^{(b-1)}\left(\frac{1}{n^{(1-\frac{b}{\alpha})}\epsilon}\right)^{(b-1)}}_{D_3}.$$

In case $\epsilon > \frac{1}{N_0}$, then $RN_0\epsilon > R$, but $|\widehat{\mu}_n(p) - \mu_*| \leq R$, that leads to D = 0. The inequality is trivially correct.

In case $\epsilon \leq \frac{1}{N_0}$, because $n < N_0$ and $b > 2, b < \alpha$, so that $0 < (1-\frac{b}{\alpha}) < 1$. Therefore $n^{(1-\frac{b}{\alpha})} < n < N_0$. Therefore, $n^{(1-\frac{b}{\alpha})}\epsilon < 1$. So that $\left(\frac{1}{n^{(1-\frac{b}{\alpha})}\epsilon}\right)^{b-1} > 1$. We can choose a constant $c' > 0$ that $D_3 > 1$, so the inequality is trivially correct.

Furthermore,

$$\lim_{n \to \infty} |\mathbb{E}[\widehat{\mu}_n(p)] - \mu_\star| \leq \lim_{n \to \infty} \mathbb{E}[|\widehat{\mu}_n(p) - \mu_\star|] = \lim_{n \to \infty} \int_0^\infty \mathbb{P}\left(|\widehat{\mu}_n(p) - \mu_\star| \geq s\right) ds$$

$$\leq \lim_{n \to \infty} \int_0^\infty c' n^{-\alpha'} s^{-\beta'} ds \leq \lim_{n \to \infty} \int_0^{n^{-\frac{\alpha'}{\beta'}}} \mathbf{1} ds + \lim_{n \to \infty} \int_{n^{-\frac{\alpha'}{\beta'}}}^\infty c' n^{-\alpha'} s^{-\beta'} ds$$

$$= \lim_{n \to \infty} c' n^{-\alpha'}\left(s^{-\beta'+1} + C\right)\Big|_{n^{-\frac{\alpha'}{\beta'}}}^\infty = 0( \text{ we need } \beta' > 1 \to \beta > 2)$$

$\square$

# E EXPERIMENTAL SETUP AND HYPERPARAMETER SELECTION

We conduct tests with $p = 1, 2, 4, 8, 10, 16$ in SyntheticTree and plot the results. We run experiments with different exploration constants $C = 0.01, 0.1, 0.25, 0.5, 0.75, 1.0, 1.25, 1.5$ and find that for Fixed-Depth-MCTS, $C = 0.1$ yields the best performance. For Stochastic-Power-UCT and UCT, the best results are obtained with $C = 0.25$. For Power-UCT, $C = 0.5$ shows the best results. When using adaptive $\{\alpha_i\}, \{\beta_i\}, \{b_i\}$ values $i \in [0, H]$, we find that $C = 0.01$ works the best.

In FrozenLake, Taxi we show results for $p = 1, 2, 2.2$. Hyperparameter search for $C$ is performed via gridsearch: $C = 0.25, 0.5, 0.75, 1.0, 1.25, 1.5$. The best performance is achieved with $c = 1.25, 1.5, 1.0, 1.0$ for UCT, Fixed-Depth-MCTS, Stochastic-Power-UCT $p = 2$ and Stochastic-Power-UCT $p = 2.2$ respectively in FrozenLake $(4 \times 4)$, with $c = 1.5, 1.0, 0.75, 0.75$ for UCT, Fixed-Depth-MCTS, Stochastic-Power-UCT $p = 2$ and Stochastic-Power-UCT $p = 2.2$ respectively in FrozenLake $(8 \times 8)$. In Taxi, we find $c = 1.5, 1.5, 1.5, 1.0$ for UCT, Fixed-Depth-MCTS, Stochastic-Power-UCT $p = 2$ and Stochastic-Power-UCT $p = 2.2$ respectively.