# OpenReview forum: "Power Mean Estimation in Stochastic Monte-Carlo Tree Search"
_auai.org/UAI/2024/Conference — UAI 2024 poster_

### Official Review · Reviewer_rwpF · 2024-03-18

**Q2-1 Originality-Novelty:** 3
**Q2-2 Correctness-Technical Quality:** 3
**Q2-5 Clarity Of Writing:** 1

**Q1 Summary And Contributions:**

The authors proposes a new MCTS algorithm, which implements a new UCB exploration function (which is called a bonus function in the paper), and a power mean estimation for the argmax of the estimated optimal arm to play at each decision node. The application is of this algorithm is for a stochastic MCTS, which the authors claim is understudied compared to the more common application of MCTS for deterministic systems.

The key theoretical guarantee for the paper is that, with the new exploration term, the error term for the value estimate decreases with O(n^{-1/2}). Convergence guarantees are also provided. Furthermore, a plethora of experimental results are provided which support the claims

**Q2-3 Extent To Which Claims Are Supported By Evidence:**

3: Good: the main claims are supported by convincing evidence (in the form of adequate experimental evaluation, proofs, (pseudo-)code, references, assumptions).

**Q2-4 Reproducibility:**

3: Good: key resources (e.g. proofs, code, data) are available and key details (e.g. proofs, experimental setup) are sufficiently well-described for competent researchers to confidently reproduce the main results.

**Q3 Main Strengths:**

The area of study, which is the problem setting pertaining to the application of MCTS to stochastic MDP setting is well justified. The strengths of this paper lies in its theoretical guarantees, much work is done to rigorously prove the claims of the paper. The experimental results are comprehensive and convincing to the reader to reinforce the theory presented.

**Q4 Main Weakness:**

Glaringly, the major weakness is the amount of grammatical and punctuation errors littered all throughout the paper. There are simply way too many, and more care needs to be taken to correct them. For example, Section 5.2, why is the word “Non-stationary” randomly capitalized? In Section 5.2 Lemma 1, it should be plural “sequence of estimators” and not “estimator”. In Remark 2, why is there is a random comma “,” hanging out in the second paragraph? In the Introduction why is are the words “years agoSilver” stuck together? In Section 4.2 why does it say “the distribution is Dirac in V_0”, rather be formal and say “the distribution is “a Dirac delta function with the domain on V_0”? W.r.t formatting, the Algorithm box is Algorithm 1 has overlapping borders with the text.

These are just some of the small English and typographical errors in the article that really affect presentation of the paper, and many more exist. The authors are strongly encouraged to do a thorough check of the grammar and improve the use of language in the paper.

Furthermore, I am not a big fan of the excessive mathematical notation used. For example the \hat{Q} estimate has an superscript h in brackets, followed by “T_s^t,a(t)” which itself is a subscript, so we essentially have a term with both superscript and subscript, and subscript within subscripts, which is menacingly confusing. In general, for the entirety of the paper, there should really be a simplification on notation (or better choice of variables), because the current version makes it very hard to follow.

**Q5 Detailed Comments To The Authors:**

I have a few questions for the authors, which they may feel free to respond to,

1. It seems like the theoretical guarantee presented in Theorem 3 applies only at the root node, correct? Would an application of this bound to any node on the tree be derivable without any major loss in generality?

2. How restrictive are the algorithmic constants in Table 1? Seems like the algorithm would only converge when those constraints are met w.r.t. \alpha, \beta and p? Do these constraints cover the majority of cases for an stochastic-MCTS in general, or are there gaps in the hyperparameter space, where convergence and value estimation bounds cannot be guaranteed?

3. Was the selection of a new exploration (bonus) term, and power mean estimation selected only with convenience to obtain theoretical guarantees, or because it presents genuine advantages for stochastic control, perhaps in specific domains. Because the results in Figure 2, doesn’t show much of a difference, if any, between Power-UCT when compared to UCT.

If the authors would kindly answer these questions, it would be greatly appreciated.

**Q9 Complying With Reviewing Instructions:**

Yes

---

> ### Author Rebuttal · Authors · 2024-04-09
>
> >Q4 Main Weakness
>
> We would like to thank the reviewer for all the suggestions, and we will take great care in the final camera version to reflect all the comments
>
>  >It seems like the theoretical guarantee presented in Theorem 3 applies only at the root node, correct? Would an application of this bound to any node on the tree be derivable without any major loss in generality?
>
> While Theorem 3 currently specifies the theoretical guarantee at the root node, we can ensure this result to any node within the tree without sacrificing generality. Through Monte-Carlo sampling in the tree search, the convergence of value estimation is ensured at every node, not solely the root node. it is worth noting that ensuring convergence guarantee at the root note is also the typical goal for planning, where the root of the tree is the current state of the world from where the planning is done.
>
> >How restrictive are the algorithmic constants in Table 1? Seems like the algorithm would only converge when those constraints are met w.r.t. $\alpha, \beta$ and $p$? Do these constraints cover the majority of cases for an stochastic-MCTS in general, or are there gaps in the hyperparameter space, where convergence and value estimation bounds cannot be guaranteed?
>
> The algorithmic constants listed in Table 1 are indeed restrictive but comprehensive, encompassing all cases of a stochastic MCTS using any power mean estimator with the constant bonus term. In experiments, $\alpha, \beta$ and $p$ have to satisfy all the conditions in Table 1 restrictively, and consider the exploration constant $C$ as a hyperparameter and try to find the best performance with a suitable value of $C$.
>
>  > Was the selection of a new exploration (bonus) term, and power mean estimation selected only with convenience to obtain theoretical guarantees, or because it presents genuine advantages for stochastic control, perhaps in specific domains. Because the results in Figure 2, doesn’t show much of a difference, if any, between Power-UCT when compared to UCT.
>
> Given the incompleteness of the theoretical analysis of UCT due to an error introduction of the estimated "logarithmic" bonus term for action selection in the tree. The theoretical understanding of our paper is indeed important, especially in stochastic settings. Moreover, the adoption of power mean estimation and a new exploration term wasn't solely for the sake of theoretical guarantees; rather, it addresses practical issues encountered in Monte-Carlo Planning, particularly concerning the underestimation of average backup and overestimation of max backup operators in UCT [1,2].
>
> The advantages of stochastic-Power-UCT include:
>
> - The use of power mean backup operators effectively balances the underestimation and overestimation in UCT[1,2].
> - The incorporation of a new exploration term guarantees the theoretical convergence when employing power mean backup in the stochastic setting.
>
> [1] "Generalized mean estimation in Monte-Carlo tree search," presented at IJCAI'20: Proceedings of the Twenty-Ninth International Joint Conference on Artificial Intelligence, January 2021, Article No.: 332, Pages 2397–2404.
>
> [2] Coulom, Rémi. "Efficient selectivity and backup operators in Monte-Carlo tree search." International conference on computers and games. Berlin, Heidelberg: Springer Berlin Heidelberg, 2006.

---

### Official Review · Reviewer_svpB · 2024-03-20

**Q2-1 Originality-Novelty:** 3
**Q2-2 Correctness-Technical Quality:** 3
**Q2-5 Clarity Of Writing:** 3

**Q1 Summary And Contributions:**

This paper presents Stochastic-Power-UCT, which employs power mean for value estimation and integrates a polynomial exploration bonus term. It marks the first endeavor with a comprehensive theoretical convergence guarantee in Stochastic MCTS. The authors establish that the estimated value function at the root node of the tree converges polynomially to the optimal value at a rate of O(n^{-1/2}). Empirical investigations on synthetic trees validate the theoretical assertions.

**Q2-3 Extent To Which Claims Are Supported By Evidence:**

3: Good: the main claims are supported by convincing evidence (in the form of adequate experimental evaluation, proofs, (pseudo-)code, references, assumptions).

**Q2-4 Reproducibility:**

3: Good: key resources (e.g. proofs, code, data) are available and key details (e.g. proofs, experimental setup) are sufficiently well-described for competent researchers to confidently reproduce the main results.

**Q3 Main Strengths:**

This paper introduces Stochastic-Power-UCT, an MCTS algorithm tailored for stochastic MDPs, employing power mean as the value estimator. The contribution extends to a comprehensive theoretical examination, establishing a convergence rate of O(n^{-1/2}). This work represents the first endeavor to offer theoretical analysis in stochastic MDPs. The paper demonstrates high technical quality, providing rigorous theoretical analysis and proofs for the convergence rate of the expected payoff.
In summary, this paper exhibits outstanding organization and clarity. It offers comprehensive definitions, derivations, proofs of Lemmas and Theorems, along with algorithms, in both the main body and appendix sections.
This paper outlines the data generation process for the synthetic tree, describes the experiment setup, and presents detailed results. However, the code associated with the research has not been made available.

**Q4 Main Weakness:**

The experiments primarily aim to showcase the numerical superiority of Stochastic-Power-UCT over UCT in the Synthetic Tree environment. However, these simulations are conducted on a relatively small scale. While the focus of the paper is primarily theoretical, expanding the evaluation to more realistic and larger-scale datasets and environments could enhance its impact. In Fig. 2 Taxi, it appears that Power-UCT and UCT exhibit no significant difference in terms of discounted returns.

**Q5 Detailed Comments To The Authors:**

Presentation: The extensive details of the proof of Lemma 1 spanned more than one page; relocating the proof to the appendix and providing additional discussion about the Lemma in the main body could enhance clarity. Question: Could you provide an explanation for the drop in return observed for Power-UCT, with p=4, in Fig. 2 FrozenLake?

**Q9 Complying With Reviewing Instructions:**

Yes

---

> ### Author Rebuttal · Authors · 2024-04-09
>
> >This paper presents Stochastic-Power-UCT, which employs power mean for value estimation and integrates a polynomial exploration bonus term. It marks the first endeavor with a comprehensive theoretical convergence guarantee in Stochastic MCTS. The authors establish that the estimated value function at the root node of the tree converges polynomially to the optimal value at a rate of O(n^{-1/2}). Empirical investigations on synthetic trees validate the theoretical assertions.
>
> We would like to thank the reviewer for acknowledging the strength and contributions of the paper
>
> >However, the code associated with the research has not been made available.
>
> We apologize for this, and we will make the experimental code available in the camera-ready version.
>
> >While the focus of the paper is primarily theoretical, expanding the evaluation to more realistic and larger-scale datasets and environments could enhance its impact.
>
> Even the Synthetic Tree environment is a toy-task that aims to evaluate the value function error at the root node, we note that the experimental scale being conducted in the paper is indeed relevant as we cover all combinations of various branching factors k= {2,4,6,8,10,12,14,16} and depths d={1,2,3,4}, given that the synthetic tree is exponentially expanded with respect to the depth of the tree, therefore increasing the complexity and stochasticity of the empirical evaluations.
>
> In Fig. 2 Taxi, it appears that Power-UCT and UCT exhibit no significant difference in terms of discounted returns.
>
> We note that this is indeed confirming the theoretical study of our approach in practice.
>
> >Presentation: The extensive details of the proof of Lemma 1 spanned more than one page; relocating the proof to the appendix and providing additional discussion about the Lemma in the main body could enhance clarity. Question: Could you provide an explanation for the drop in return observed for Power-UCT, with p=4, in Fig. 2 FrozenLake?
>
> We thank the reviewer for pointing this out. In the final camera-ready version, we commit to address all noted issues regarding notations and theoretical presentation, including the relocation of Lemma 1, with utmost diligence.
> Regarding the observed drop in the return of Power-UCT with
> p=4 in Figure 2 (FrozenLake), we attribute this to the circumstance where the average of discounted rewards falls within the confidence interval. To mitigate this, we enhanced the robustness of our results by increasing the number of random seeds to 1000 in the final version (results shown in Table below), thereby reducing the impact of the confidence interval.
>
> Discounted total reward in FrozenLake for the comparison methods. Mean $\pm$ 2 times standard deviation error are computed from $1000$ simulations.  Bold denotes no statistically significant difference to the highest mean (t-test, $p < 0.05$)
>
> \begin{array}{|l|c|c|c|c|c|}
> \text{Algorithms} & 2^{8} samples & 2^{10} samples & 2^{12} samples & 2^{14} samples & 2^{16} samples \\\\
> \text{UCT} & 0.05 \pm 0.014 & \mathbf{0.08 \pm 0.016} & \mathbf{0.14 \pm 0.020} & 0.28 \pm 0.024 & 0.43 \pm 0.023\\\\
> p=2 & \mathbf{0.06 \pm 0.016} & \mathbf{0.10 \pm 0.017} & \mathbf{0.21 \pm 0.022} & \mathbf{0.37 \pm 0.024} & \mathbf{0.44 \pm 0.023}\\\\
> p=2.2 & \mathbf{0.07 \pm 0.017} & \mathbf{0.12 \pm 0.019} & \mathbf{0.21 \pm 0.022} & \mathbf{0.37 \pm 0.024} & \mathbf{0.44 \pm 0.023}
> \end{array}
>
> >However, these simulations are conducted on a relatively small scale.
>
> We thank the reviewer for pointing this out. We think that small-scale environments enable a more in-depth understanding of the practical advantage of the proposed method, and scaling up to some real-world, practical settings would be of interest in future works.

---

### Official Review · Reviewer_vbD7 · 2024-03-23

**Q2-1 Originality-Novelty:** 3
**Q2-2 Correctness-Technical Quality:** 3
**Q2-5 Clarity Of Writing:** 3

**Q10 Ethical Concerns:**

None.

**Q1 Summary And Contributions:**

The paper proposes a new MCTS algorithm that uses a newly introduced Stochastic-Power-UCT criterion for choosing the arm to pull.  Stochastic-Power-UCT is motivated by UCT, theoretically extending UCT to account for stochasticity as well as designed more generally for the power mean value function, with the more common average mean as a special case.

The paper asserts that the new Stochastic-Power-UCT method exhibits the same convergence rate of $O(n^{1/2})$ as UCT.

Experiments are performed on various stochastic MDP's (from existing literature/prior work) and show better performance compared to UCT.

**Q2-3 Extent To Which Claims Are Supported By Evidence:**

3: Good: the main claims are supported by convincing evidence (in the form of adequate experimental evaluation, proofs, (pseudo-)code, references, assumptions).

**Q2-4 Reproducibility:**

3: Good: key resources (e.g. proofs, code, data) are available and key details (e.g. proofs, experimental setup) are sufficiently well-described for competent researchers to confidently reproduce the main results.

**Q3 Main Strengths:**

* Despite being dense, the paper is well written and contributes a non-trivial extension to the well known and used UCT algorithm for the stochastic case and for use with a power-mean estimator.
* The authors include and extensive theoretical justification for their claims
* Experiments are conducted on a wide array of benchmarks adopted from existing works

**Q4 Main Weakness:**

* The paper is quite dense and could benefit from including various levels of description/justification to allow for a wider audience
* Parameterization used for the proposed schemed during experimentation seems ad-hoc
* Only UCT is compared with

**Q5 Detailed Comments To The Authors:**

* Being more of a theoretically driven work, the paper is quite dense.  It would be helpful to...
    * highlight certain important results (ie. help key points stand out in the sea of equations)
    * supplement with more intuition
1.  What was the reason to use no discounting in your experiments?
2.  What is your understanding about the parameterization for your method?  Is there currently any intuition about what hyperparameters to use apriori?
3.  Intuitively, how would you explain the better performance you see by your method as compared to UCT?
4.  Would it make sense to also compare with other methods?

**Q9 Complying With Reviewing Instructions:**

Yes

---

> ### Author Rebuttal · Authors · 2024-04-09
>
> > Main Strengths:
> Despite being dense, the paper is well written and contributes a non-trivial extension to the well known and used UCT algorithm for the stochastic case and for use with a power-mean estimator.
> The authors include and extensive theoretical justification for their claims
> Experiments are conducted on a wide array of benchmarks adopted from existing works
>
> We thank the reviewer for acknowledging the main strengths of the paper
>
> >The paper is quite dense and could benefit from including various levels of description/justification to allow for a wider audience
> Parameterization used for the proposed scheme during experimentation seems ad-hoc
>
> We apologize for that, and as promised, we will take serious considerations to improve the theoretical presentation with fixed notions of the paper.
>
> >Only UCT is compared with
>
> While the comparison of Power-UCT with other methods has been previously explored [1], we acknowledge the significance of conducting a comprehensive comparison with alternative approaches to demonstrate the advantages of our paper. We intend to include this comparison in the final camera-ready version.
>
> >Being more of a theoretically driven work, the paper is quite dense. It would be helpful to...highlight certain important results (ie. help key points stand out in the sea of equations) supplement with more intuition
>
> we apologize for the confusion regarding the presentation of the paper, and we will take great care in the final camera-ready version.
>
> >What was the reason to use no discounting in your experiments?
>
> In FrozenLake, NChain, Taxi we set the discount factor \gamma = 0.99. In Synthetic Tree, since the depth of the tree is only 1,2,3,4 for convenience, we set \gamma = 1
>
> >What is your understanding about the parameterization for your method? Is there currently any intuition about what
> hyperparameters to use apriori?
>
> All the parameters have to restrictively satisfy all the conditions in Table 1. The exploration constant $C$ is considered as a hyperparameter.
>
> >Intuitively, how would you explain the better performance you see by your method as compared to UCT?
>
> The advantage in performance of Stochastic-Power-UCT can be explained as the use of the power mean offers a practical solution to the issue of underestimation of the average mean and overestimation of the max backup operator in UCT [1,2] making it a more balanced estimation of values, effectively mitigating the problem of biased estimation. Stochastic-Power-UCT further ensures the convergence guarantee, rendering it a robust choice for stochastic planning tasks.
>
> >Would it make sense to also compare with other methods?
>
> In our experimental setup, our primary goal was to evaluate the performance of Stochastic-Power-UCT specifically in comparison to UCT, since our method is for stochastic environments. It is worth noting that previous research has extensively investigated the comparison of Deterministic Power-UCT with other methods [1], so it was not considered in our study. However, we acknowledge the reviewer's valid point about the need for completeness and agree that including a numerical comparison with Deterministic Power-UCT in the final version of our paper, possibly in the appendix, would provide a more comprehensive analysis. We commit to addressing this suggestion in the camera-ready version to improve the thoroughness of our study.
>
> [1] Reference: "Generalized mean estimation in Monte-Carlo tree search," presented at IJCAI'20: Proceedings of the Twenty-Ninth International Joint Conference on Artificial Intelligence, January 2021, Article No.: 332, Pages 2397–2404.
> [2] Coulom, Rémi. "Efficient selectivity and backup operators in Monte-Carlo tree search." International conference on computers and games. Berlin, Heidelberg: Springer Berlin Heidelberg, 2006.

---

### Official Review · Reviewer_kZ1U · 2024-03-25

**Q2-1 Originality-Novelty:** 2
**Q2-2 Correctness-Technical Quality:** 3
**Q2-5 Clarity Of Writing:** 3

**Q1 Summary And Contributions:**

This paper studies Monte-Carlo Tree Search, and extends previous results of using polynomial exploration bonuses to stochastic Markov Decision Processes (MDPs). The proposed Stochastic-Power-UCT method uses power mean as the value estimator, and the authors show that it shares the same convergence rate $O(n^{-1/2})$ as UCT. Empirical results on several environments (SyntheticTree, FrozenLake, NChain, Taxi) show that the proposed Stochastic-Power-UCT method achieves comparable or better performances than UCT.

**Q2-3 Extent To Which Claims Are Supported By Evidence:**

2: Fair: the main claims are somewhat supported by evidence (but the experimental evaluation may be weak, or does not match entirely with the claims, important baselines may be missing, proofs contain important ideas but lack rigor, algorithmic details are only discussed superficially, references are imprecise, assumptions are not sufficiently motivated or explicated, etc.).

**Q2-4 Reproducibility:**

3: Good: key resources (e.g. proofs, code, data) are available and key details (e.g. proofs, experimental setup) are sufficiently well-described for competent researchers to confidently reproduce the main results.

**Q3 Main Strengths:**

1. Improving over the UCT and other MCTS methods is a relevant problem.
2. Experiments verify the proposed methods.

**Q4 Main Weakness:**

1. Extending existing results from deterministic to stochastic settings has a point but also limits the technical novelty.
2. The technical contributions could possibly be presented in a better way. Despite the theorems, it is confusing to me why power mean has to be used in MCTS and how it overcomes the issues in UCT.
3. Experiments are on simple tasks, and tree search algorithms have been used in more complicated environments. To argue that the proposed method can replace UCT, more experimental results are needed.

**Q5 Detailed Comments To The Authors:**

1. If we increase $p$ to larger and larger values, how the power mean would behave in its limit?
2. What is a good choice of $p$ value and why is that.
3. What would be a good intuition of using power mean and how is it a better quantity to estimate?
4. How do you compare the convergence rate of Stochastic-Power-UCT and UCT, given they are both $O(n^{-1/2})$ (better constants?).

**Q9 Complying With Reviewing Instructions:**

Yes

---

> ### Author Rebuttal · Authors · 2024-04-09
>
> >Extending existing results from deterministic to stochastic settings has a point but also limits the technical novelty.
>
> We would like to emphasize that extending existing results from deterministic to stochastic settings represents a significant advancement rather than a mere extension. While MCTS has found successful applications in various domains such as video games, robotics, autonomous car driving, high level action planning tasks, the theoretical understanding of MCTS remains limited. Moreover, most real world scenarios are stochastic, underscoring the importance of theoretical understanding in stochastic environments. Therefore, the extension to stochastic settings is not only relevant but also crucial for advancing the applicability and effectiveness of MCTS in practical contexts.
>
> >Despite the theorems, it is confusing to me why power mean has to be used in MCTS and how it overcomes the issues in UCT.
>
> As mentioned before, The power mean backup operator used in Power-UCT solves underestimation/overestimation issues in UCT[1,2].
>
> >If we increase p to larger and larger values, how the power mean would behave in its limit?
>
> when $p= 1$, the power mean backup operator becomes the average mean in UCT. On the other hand, when $p= +\infty$, we get to the max backup operator. In experiments, we increase the value of $p$ to help trade-off between average and max and find the best performance. As in the SyntheticTree experiments, when increasing the value of $p$, the performance increases with $p = 1, 2, 4, 8$ and decreases with $p = 16$ in SyntheticTree with (k=16, d=4).
>
> >What is a good choice of p  value and why is that.
>
> In SyntheticTree, FrozenLake, NChain, and Taxi, we find that the performance is better or at least the same with p = 2 compared to UCT across all environments.
>
> >What would be a good intuition of using power mean and how is it a better quantity to estimate?
>
> Using the power mean provides a valuable solution to the problem of underestimation of the average mean and overestimation of the max backup operator in UCT[1]. By using the power mean as a value backup operator, Stochastic-Power-UCT achieves a more balanced value estimate, effectively addressing the problem of biased estimation. This adjustment not only ensures accurate estimation but also maintains the convergence guarantee, making it a robust choice for stochastic planning tasks.
> [1] ''Generalized mean estimation in Monte-Carlo tree search,'' presented at IJCAI'20: Proceedings of the Twenty-Ninth International Joint Conference on Artificial Intelligence, January 2021, Article No.: 332, Pages 2397–2404.
>
> >How do you compare the convergence rate of Stochastic-Power-UCT and UCT, given they are both $\mathcal{O}(n^{-1/2})$ (better constants)?
>
> We would like to emphasize that Stochastic-Power-UCT, by employing power mean as an estimator, shares the same convergence rate as UCT, both exhibiting a rate of $\mathcal{O}(n^{-1/2})$. Consequently, UCT can be regarded as a special case of Stochastic-Power-UCT. Moreover, particularly when p=2, Stochastic-Power-UCT consistently outperforms UCT, underscoring its significant role in the field. This is especially noteworthy given the widespread applications of Monte Carlo Planning in stochastic environments.
>
> Stochastic-Power-UCT and UCT share the same convergence rate which is an interesting finding.
>
> [1] "Generalized mean estimation in Monte-Carlo tree search," presented at IJCAI'20: Proceedings of the Twenty-Ninth International Joint Conference on Artificial Intelligence, January 2021, Article No.: 332, Pages 2397–2404.
>
> [2] Coulom, Rémi. "Efficient selectivity and backup operators in Monte-Carlo tree search." International conference on computers and games. Berlin, Heidelberg: Springer Berlin Heidelberg, 2006.

---

### Official Review · Reviewer_TbYD · 2024-03-25

**Q2-1 Originality-Novelty:** 2
**Q2-2 Correctness-Technical Quality:** 2
**Q2-5 Clarity Of Writing:** 1

**Q1 Summary And Contributions:**

This paper describes a theoretical analysis of Monte Carlo tree search (MCTS) with stochastic Markov decision processes (MDPs).  Namely, the Power-UCT algorithm is extended to stochastic environments, resulting in the Stochastic-Power-UCT algorithm.  A polynomial convergence rate is established, and empirical results in a synthetic problem and several MDP benchmarks demonstrate the performance of the algorithm.

**Q2-3 Extent To Which Claims Are Supported By Evidence:**

3: Good: the main claims are supported by convincing evidence (in the form of adequate experimental evaluation, proofs, (pseudo-)code, references, assumptions).

**Q2-4 Reproducibility:**

3: Good: key resources (e.g. proofs, code, data) are available and key details (e.g. proofs, experimental setup) are sufficiently well-described for competent researchers to confidently reproduce the main results.

**Q3 Main Strengths:**

S1) A complete theoretical understanding of MCTS under stochastic state transitions is important, considering how many real-world environments have stochastic transitions instead of deterministic.  I expect the work to be of interest to the planning and reinforcement learning communities at UAI.

S2) I appreciated the inclusion of both theoretical and empirical evaluation.  The use of common benchmarks like Frozen Lake and Taxi aid the reader who is likely familiar with these problems.

**Q4 Main Weakness:**

W1) The mathematical notation is very inconsistent and difficult to follow.  This greatly hampered my ability to follow the theoretical analysis.  Examples are provided below.

W2) It was unclear what exactly was novel in Stochastic-Power-UCT when compared to Power-UCT.  The only difference I could tell was the heuristic used in SelectAction (as a result of the theoretical analysis).

W3) I was surprised to find only a comparison to UCT in the experiments and not the deterministic Power-UCT algorithm.  This made it difficult to assess whether the difference in performance was due to the difference in how the Q values are are calculated vs. the novel heuristic for action selection.

W4) The experimental results lacked any statistical analysis to determine whether the differences in performance were statistically significant or not.  Standard deviations are not enough to determine whether one approach outperformed another.

**Q5 Detailed Comments To The Authors:**

The mathematical notation really made this paper difficult to follow.  Much of the problem was

1) renaming of common variables from the majority of the MCTS literature applied to MDPs (e.g., replacing the number of times a node is visited with T_{s, a}(t))

2) incorrect definitions.  For example, the bonus of vanilla UCT is incorrectly defined as C \sqrt{\frac{log(t)}{s}}, but s is a state and t should the visit count and not trajectory number (like t is used everywhere else).  Also, the transition function defined on page 2 is P: S x A \rightarrow S is deterministic, but the focus of the paper is on stochastic transitions.

3) being inconsistent in how subscripts and function parameters are used.  For example, the visit counts T use state and action as subscripts and trajectory number as a function argument, whereas Q and V use counts as subscripts and states (and actions) as function arguments.

Also missing from the discussion is how similar the UCB-1 bonus of c \sqrt{\frac{log n(s)}{n(s, a}} is to your bonus in Remark 2 of c \sqrt{\frac{\sqrt{n(s)}}{n(s, a)}}.  The only difference is UCB-1 has log n(s) in the numerator, whereas you have \sqrt{n(s)}.  Visualized, the two are pretty similar, but asymptotically, log(n) < \sqrt(n) as n increases, so considering stochastic transitions increases the amount of exploration bonus required (which makes sense).

**Q9 Complying With Reviewing Instructions:**

Yes

---

> ### Author Rebuttal · Authors · 2024-04-09
>
> >the main claims are supported by convincing evidence (in the form of adequate experimental evaluation, proofs, (pseudo-)code, references, assumptions).
>
> We thank you for the positive comments.
>
> > The mathematical notation is very inconsistent and difficult to follow. This greatly hampered my ability to follow the theoretical analysis. Examples are provided below.
>
> We greatly apology for the inconsistent and cumbersome notations. We will take special care to fix and lighten all the notations in the final camera version.
>
> >It was unclear what exactly was novel in Stochastic-Power-UCT when compared to Power-UCT. The only difference I could tell was the heuristic used in SelectAction (as a result of the theoretical analysis).
>
> Stochastic-Power-UCT is designed for stochastic settings while Power-UCT is originated for deterministic.
>
> >I was surprised to find only a comparison to UCT in the experiments and not the deterministic Power-UCT algorithm. This made it difficult to assess whether the difference in performance was due to the difference in how the Q values are calculated vs. the novel heuristic for action selection.
>
> In our experiments, the focus was specifically on comparing Stochastic-Power-UCT to UCT, given the novelty of the method in stochastic environments. It's important to note that the comparison of Deterministic Power-UCT with other methods has already been extensively studied in prior literature[1], which is why it wasn't included in our experiments. However, we acknowledge the reviewer's point regarding completeness and agree that adding a numerical comparison with Deterministic Power-UCT in the camera-ready version would provide a more comprehensive analysis. We will ensure to include this comparison, at least in the appendix, to address the reviewer's concern.
>
> [1] "Generalized mean estimation in Monte-Carlo tree search," presented at IJCAI'20: Proceedings of the Twenty-Ninth International Joint Conference on Artificial Intelligence, January 2021, Article No.: 332, Pages 2397–2404.
>
> >The experimental results lacked any statistical analysis to determine whether the differences in performance were statistically significant or not. Standard deviations are not enough to determine whether one approach outperformed another.
>
> We run $1000$ random seeds in FrozenLake and show the performance as shown in the Table below. In the final camera-ready version, we will present the results obtained from running simulations with 1000 random seeds for FrozenLake, NChain, and Taxi environments. Additionally, we will conduct statistical significance tests to provide robust validation of our findings. This is an easy addition.
>
> Discounted total reward in FrozenLake for the comparison methods. Mean $\pm$ 2 times standard deviation error are computed from $1000$ simulations.  Bold denotes no statistically significant difference to the highest mean (t-test, $p < 0.05$)
>
>
> \begin{array}{|l|c|c|c|c|c|}
> \text{Algorithms} & 2^{8} samples & 2^{10} samples & 2^{12} samples & 2^{14} samples & 2^{16} samples \\\\
> \text{UCT} & 0.05 \pm 0.014 & \mathbf{0.08 \pm 0.016} & \mathbf{0.14 \pm 0.020} & 0.28 \pm 0.024 & 0.43 \pm 0.023\\\\
> p=2 & \mathbf{0.06 \pm 0.016} & \mathbf{0.10 \pm 0.017} & \mathbf{0.21 \pm 0.022} & \mathbf{0.37 \pm 0.024} & \mathbf{0.44 \pm 0.023}\\\\
> p=2.2 & \mathbf{0.07 \pm 0.017} & \mathbf{0.12 \pm 0.019} & \mathbf{0.21 \pm 0.022} & \mathbf{0.37 \pm 0.024} & \mathbf{0.44 \pm 0.023}
> \end{array}

---

### Meta-Review · Area_Chair_zUcY · 2024-04-23

The paper provides a theoretical analysis of Monte Carlo Tree Search (MCTS) for stochastic domains.  This represents an important theoretical advance.  However the presentation is weak.  As pointed out by the reviewers, the notation is inconsistent and there are many issues with the language.  Nevertheless, this is good technical work.